# Comparison of secondary organic aerosol formation from toluene on initially wet and dry ammonium sulfate particles at moderate relative humidity

Tengyu Liu[1], Dan Dan Huang[1], Zijun Li[2], Qianyun Liu[3], ManNin Chan[2,4], and Chak K. Chan[1,*]

1. School of Energy and Environment, City University of Hong Kong, Hong Kong, China

2. Earth System Science Programme, The Chinese University of Hong Kong, Hong Kong, China

3. Division of Environment and Sustainability, Hong Kong University of Science and Technology, Hong Kong, China

4. The Institute of Environment, Energy and Sustainability, The Chinese University of Hong Kong, Hong Kong, China

*Corresponding author:

Chak K. Chan

School of Energy and Environment, City University of Hong Kong, China

Tel: +852-34425593

Email: Chak.K.Chan@cityu.edu.hk

## Abstract

The formation of secondary organic aerosol (SOA) has been widely studied in the presence of dry seed particles at low relative humidity (RH). At higher RH, initially dry seed particles can exist as wet particles due to water uptake by the seeds as well as the SOA. Here, we investigated the formation of SOA from the photooxidation of toluene using an oxidation flow reactor in the absence of $NO_x$ under a range of OH exposures on initially wet or dry ammonium sulfate (AS) seed particles at an RH of 68%. The ratio of the SOA yield on wet AS seeds to that on dry AS seeds, the relative SOA yield, decreased from $1.31\pm0.02$ at an OH exposure of $4.66\times10^{10}$ molecules $cm^{-3}$ s to $1.01\pm0.01$ at an OH exposure of $5.28\times10^{11}$ molecules $cm^{-3}$ s. This decrease may be due to the early deliquescence of initially dry AS seeds after coated by highly oxidized toluene-derived SOA. SOA formation lowered the deliquescence RH of AS and resulted in the uptake of water by both AS and SOA. Hence the initially dry AS seeds contained aerosol liquid water (ALW) soon after SOA formed and the SOA yield and ALW approached those of the initially wet AS seeds as OH exposure and ALW increased, especially at high OH exposure. However, a higher oxidation state of the SOA on initially wet AS seeds than that on dry AS seeds was observed at all levels of OH exposure. The difference in mass fractions of *m/z* 29, 43 and 44 of SOA mass spectra, obtained using an aerosol mass spectrometer (AMS), indicated that SOA formed on initially wet seeds may be enriched in earlier-generation products containing carbonyl functional groups at low OH exposures and later-generation products containing acidic functional groups at high exposures. Our results suggest that

inorganic dry seeds become at least partially deliquesced particles during SOA
formation and hence ALW is inevitably involved in the SOA formation at moderate RH.
More laboratory experiments conducted with a wide variety of SOA precursors and
inorganic seeds under different $NO_x$ and RH conditions are warranted.

## 1. Introduction

Secondary organic aerosol (SOA) is an important component of atmospheric particulate matter, which influences air quality, climate and human health (Hallquist et al., 2009). SOA is mainly formed via the oxidation of volatile organic compounds (VOCs), followed by partitioning to the condensed phase. Traditional atmospheric chemical transport models largely underestimate the levels of SOA (de Gouw et al., 2005; Volkamer et al., 2006; Hodzic et al., 2010) and the degree of oxidation (Rudich et al., 2007; Ng et al., 2010). The updated models incorporating the volatility basis set (VBS) formalism (Donahue et al., 2006) can better predict the observed SOA, but SOA formation still remains under-constrained (Shrivastava et al., 2011; Tsigaridis et al., 2014; Hayes et al., 2015; Ma et al., 2017). SOA yields in atmospheric chemical transport models are obtained from smog chamber experiments using dry seed particles (Barsanti et al., 2013; Mahmud and Barsanti, 2013) under dry conditions. Yet, atmospheric relative humidity is often sufficiently high that aerosols often contain aerosol liquid water (ALW) due to their hygroscopic properties (Liao and Seinfeld, 2005; Lee and Adams, 2010; Guo et al., 2015; Nguyen et al., 2016). The presence of ALW in aerosols may enhance SOA formation by facilitating the partitioning of semivolatile organic compounds and the uptake of water-soluble gases through aqueous-phase reactions (Hennigan et al., 2008; Lim et al., 2010; Ervens et al., 2011; Lee et al., 2011; Sareen et al., 2017). ALW may also promote photodegradation of dissolved SOA (Romonosky et al., 2014). Therefore, SOA formation under atmospherically relevant relative humidity needs to be better constrained in

atmospheric chemical transport models by incorporating ALW. In addition, understanding water uptake of SOA is important for estimating its loss by wet deposition, which is not well constrained.

Aromatic hydrocarbons constitute a large fraction of the total non-methane hydrocarbons in the urban atmosphere (Calvert et al., 2002) and account for a significant fraction of SOA in urban areas (Ding et al., 2012; Zhao et al., 2017). Toluene is the most abundant aromatic hydrocarbon (Calvert et al., 2002; Zhang et al., 2016) and SOA yields from the photooxidation of toluene on dry or wet ammonium sulfate (AS) seeds has been studied by varying the RH in smog chambers. Kamens et al. (2011) observed higher yields of SOA from toluene at higher RHs. They attributed this increase to the initially wet seed particles. On the other hand, Edney et al. (2000) reported that wet seeds had no effect on the SOA yields of toluene compared with dry seeds. In these studies, different RHs used for dry and wet seeds experiments may influence the gas-phase chemistry and complicate the comparison of SOA formation.

SOA formation on initially dry and wet AS seeds has been compared using oxidation flow reactors at same RHs (Wong et al., 2015; Faust et al., 2017). Faust et al. (2017) found a 19% enhancement in the SOA yield of toluene on wet AS seeds over that on dry AS seeds at 70% RH. However, at such high RH, the initially dry and water-free AS seed particles can uptake water upon SOA formation because SOA themselves can be hygroscopic and they can also lower the deliquescence RH of the AS seeds (Takahama et al., 2007; Smith et al., 2011, 2012, 2013). The potential influence of SOA formation on the physical state of the initially dry seeds as well as and the overall water

uptake by the aged particles was not explicitly discussed. In addition, the hydroxyl
radicals (OH) exposure in Faust et al. (2017) was approximately $2\times10^{11}$ molecules cm$^{-}$
$^{3}$ s, equivalent to about 1.5 days of oxidation in the atmosphere assuming an ambient
OH concentration of $1.5\times10^{6}$ molecules cm$^{-3}$ (Mao et al., 2009). Atmospheric particles
can undergo oxidation for as long as 1-2 weeks (Balkanski et al., 1993).
In this study, SOA formation from the photooxidation of toluene was investigated
in an oxidation flow reactor at an RH of 68% under a wide range of OH exposures using
initially wet or dry AS seed particles. The yields and composition of SOA as well as the
estimated ALW contents for the initially wet and dry seeds are compared. We found
that as OH exposure increased, the SOA yield and ALW of the initially dry seeds
approached those of the initially wet seeds while the wet seeds yielded SOA of a higher
degree of oxidation than the dry seeds did at all exposure levels.

## 2. Materials and methods

### 2.1 Generation of seed particles

A schematic of the experimental setup, similar to that used in Wong et al. (2015) and
Faust et al. (2017), is shown in Fig. 1. AS seed particles were generated from an aqueous
AS solution (Sigma-Aldrich) using an atomizer (TSI 3076, TSI Inc., USA). In
experiments using dry seeds, the atomized aqueous AS droplets passed through a silica
gel diffusion dryer so that the RH was reduced to less than 30% at which AS effloresced,
while in experiments using wet seeds, they bypassed the diffusion dryer. The dry or wet
seed particles then entered and mixed with a humidified $N_2/O_2/O_3$ flow in an oxidation
flow reactor. The RH in the flow reactor was at 68%, which lies between the

efflorescence and deliquescence RH of AS (Seinfeld and Pandis, 2006), so that the seed particles remained in their original phase with the wet particles containing ~18.6 µg m$^{-3}$ ALW (see Section 2.4) and the dry particles anhydrous before reaction started. Hereafter, the experiments using initially wet and dry AS seed particles are simplified as wet and dry AS seeds, respectively. "Wet" and "dry" refer to the initial state of the seed particles before SOA formation.

When atomizing a given AS solution, the diameter of wet AS droplets is much larger than that of dry AS particles due to the water uptake of AS (Chan et al., 1992), resulting in a larger surface area of seed particles. Previous studies have demonstrated that a large surface area of seed particles may increase the SOA yields by reducing the wall loss of organic vapors (Matsunaga and Ziemman, 2010, Zhang et al., 2014, 2015; Huang et al., 2016; Krechmer et al., 2016). To obtain seed particles of comparable surface areas, we atomized 0.013 mM and 0.015 mM of the AS solution for wet and dry AS seeds, respectively. As shown in Fig. S1, the surface area distribution of initially wet AS seeds was similar to that of initially dry AS seeds. Because of the difference in AS concentration between the stock solutions used, wet AS seeds had a mean diameter of 88 nm and were slightly smaller than dry AS seeds which had a mean diameter of 102 nm. The total surface area of wet AS seeds was 21% larger than that of dry AS seeds. The mass loading of wet and dry AS seeds was 31.0 and 24.2 µg m$^{-3}$, respectively.

**2.2 Oxidation flow reactor**

SOA formation from the photooxidation of toluene in the absence of NO$_x$ on initially dry or wet seeds was investigated in a potential aerosol mass (PAM) oxidation flow

reactor, which has been described in detail elsewhere (Kang et al., 2007, 2011; Lambe
et al., 2011a, 2015; Liu et al., 2017). Briefly, a PAM chamber is a continuous oxidation
flow reactor using high and controlled levels of oxidants to oxidize gaseous precursors
to produce SOA. The chamber used in this study had a volume of approximately 19 L
(length 60 cm, diameter 20 cm). The total flow rate in the PAM chamber was set at 3 L
$min^{-1}$ using mass flow controllers, resulting in a residence time of approximately 380 s.
The RH and temperature of the PAM outflow were measured continuously (HMP 110,
Vaisala Inc, Finland) and stabilized at approximately 68% and 20 ℃, respectively. High
OH exposures were realized through the photolysis of ozone irradiated by a UV lamp
($\lambda = 254$ nm) in the presence of water vapor. Ozone was produced by an ozone generator
(1000BT-12, ENALY, Japan) via the irradiation of pure $O_2$. The OH concentration was
adjusted by varying the concentration of ozone in the PAM chamber from 0.4 ppm to
4.3 ppm. The corresponding upper limit of OH exposure at these operating conditions
ranged from $0.47 \times 10^{11}$ molecules $cm^{-3}$ s to $5.28 \times 10^{11}$ molecules $cm^{-3}$ s, equivalent to
0.36 to 4.08 days of atmospheric oxidation assuming an ambient OH concentration of
$1.5 \times 10^6$ molecules $cm^{-3}$ (Mao et al., 2009). The upper limit of OH exposure was
determined by measuring the decay of $SO_2$ (Model T100, TAPI Inc., USA) in the
absence of toluene, following procedures described elsewhere (Kang et al., 2007;
Lambe et al., 2011a). The reduction in OH exposure due to the addition of toluene was
estimated to range from 15% at the highest OH exposure to 25% at the lowest OH
exposure, using the method of Peng et al. (2016). Peng et al. (2016) found that non-OH
chemistry, including photolysis at $\lambda = 254$ nm and reactions with $O(^1D)$, $O(^3P)$ and $O_3$,
may play an important role in oxidation flow reactors. In this study, the PAM reactor
was operated at water vapor mixing ratios above 0.5% and external OH reactivity below
20 s$^{-1}$. Non-OH chemistry is expected to play a negligible role under these conditions
(Peng et al., 2016).

Before and after each experiment, the PAM reactor was cleaned under an OH

exposure of ~$1\times10^{12}$ molecules cm$^{-3}$ s until the mass concentration of background
particles dropped below 3 μg m$^{-3}$. After characterizing dry or wet AS seed particles for
half an hour, the UV lamp was turned on to oxidize the background gases at five
different OH levels to measure the concentrations of background organics. A toluene
mixture (29.6 ppm in nitrogen) with a flow rate of 0.013 L min$^{-1}$ was then introduced
to initiate SOA formation. The initial concentration of toluene in the PAM reactor was
approximately 138 ppb. The reacted and final concentrations of toluene were calculated
from the OH exposure and the rate constant of the reaction between toluene and OH
(Atkinson and Arey, 2003) (Table 1). The flow and light conditions were the same for
initially wet and dry seeds. Therefore, the quantification of toluene would not introduce
uncertainties to the relative SOA yields described in Section 3.1 as the initial
concentrations of toluene and OH exposures were the same for both cases. SOA was
measured for at least an hour with a step-wise increase in the five OH levels.
**2.3 Characterization of non-refractory components**
The AS/SOA mixed particles were characterized for the chemical composition of non-
refractory components including organics, sulfate and ammonium as well as the
elemental ratios of organics using a high-resolution time-of-flight aerosol mass
spectrometer (hereafter AMS, Aerodyne Research Incorporated, USA) (DeCarlo et al.,
2006). The instrument was operated in the high sensitivity V-mode and the high
resolution W-mode alternating every one minute. The toolkit Squirrel 1.57I and Pika
1.16I were used to analyze the AMS data. The molar ratios of hydrogen to carbon (H:C)
and oxygen to carbon (O:C) were determined using the Aiken method (Aiken et al.,
2007, 2008). The ionization efficiency of the AMS was calibrated using 300 nm
ammonium nitrate particles. The particle-free matrix air, obtained by passing the air
flow from the PAM reactor through a HEPA filter, was measured for at least 20 min
before each experiment to determine the signals from major gases.

The collection efficiency (CE) of an AMS is dependent on the chemical

composition and acidity as well as the phase state of particles (Matthew et al., 2008;
Middlebrook et al., 2012). Matthew et al. (2008) found that the CE for solid particles
thickly coated with liquid organics was 100%. In this study, experiments were
conducted at an RH of 68%, exceeding the RH threshold for the semisolid-to-liquid
phase transition for toluene-derived SOA (Bateman et al., 2015; Song et al., 2016). The
toluene-derived SOA in these experiments was therefore liquid-like. The unimodal size
distributions of particle numbers show the SOA formation on AS seed particles without
much nucleation mode particles (Fig. S2). A CE of 1 was used for processing all AMS
data since the AS seed particles were coated by liquid SOA. The adoption of this CE
value was supported by that the concentration of sulfate measured with the AMS varied
by less than 5% of the average mass of sulfate after coated by SOA for both wet and
dry AS seeds conditions. For the quantification of SOA, the contribution from

background organic aerosols was subtracted from the total organic aerosols. The ratio

of SOA mass to background organic mass ranged from 7 to 59, indicating that the

contribution from background organics was negligible. Aerosol particles typically pass

through a silica gel diffusion dryer to remove ALW before they are measured by AMS.

However, this may lead to some losses of semivolatile organics through reversible

partitioning (Wong et al., 2015; Faust et al., 2016). In this study, the AS/SOA mixed

particles stream passed through and bypassed a diffusion dryer alternately before they

were measured by AMS. Overall less than 8% of SOA were lost for wet and dry AS

seeds after passing the diffusion dryer (Fig. S3), possibly due to reversible partitioning

of the SVOCs. In this paper, the data reported are those bypassing the diffusion dryer.

A scanning mobility particle sizer (SMPS, TSI Incorporated, USA, classifier model

3082, CPC model 3775) was used to measure particle number concentrations and size

distributions. Particle size ranged from 15 nm to 661 nm.

To evaluate the influence of seed surface area on SOA formation, we conducted

another experiment at OH exposure of $0.47 \times 10^{11}$ molecules cm$^{-3}$ s with 50% of the seed

surface area used in the wet AS experiment. The difference in SOA concentration was

approximately 1% between these two experiments. Hence the 20% difference in seed

surface area as well as the difference in mass loadings between wet and dry AS particles

cannot account for the difference in SOA yield to be discussed below.

**2.4 Estimation of aerosol liquid water (ALW) content**

The ALW content of the initially dry AS was zero. However, as reactions proceed, SOA

themselves can uptake water and also lower the deliquescence RH of AS, leading to

water uptake by AS and some fractions of AS in aqueous phase. The ALW contents of
AS (ALW$_{AS}$) and toluene-derived SOA (ALW$_{SOA}$) were estimated from the following
equations (Kreidenweis et al., 2008):

$$ALW_{AS} = V_{AS}\kappa_{AS}\, f\, \frac{\alpha_w}{1-\alpha_w}\, \rho_w \qquad (1)$$

$$ALW_{SOA} = V_{SOA}\kappa_{SOA}\, \frac{\alpha_w}{1-\alpha_w}\, \rho_w \qquad (2)$$

where $V_{AS}$ and $V_{SOA}$ represent the volume concentrations of dry AS and SOA particles,
$\kappa_{AS}$ is the hygroscopicity parameter of AS particles obtained from Kreidenweis et al.
(2008), $\kappa_{SOA}$ is the hygroscopicity parameter of toluene-derived SOA calculated using
the linear correlation between $\kappa_{SOA}$ and the O:C ratios of SOA proposed by Lambe et al.
(2011b), the term $f$ is the fraction of AS particles that dissolved, $\alpha_w$ is the water activity
and $\rho_w$ is the density of water (1.0 g cm$^{-3}$). Here, $\alpha_w$ was assumed to be equivalent to
RH/100 for simplicity. The volume concentrations of dry AS and SOA particles were
estimated from the measured mass concentration of AS and SOA assuming their
respective particle densities to be 1.77 g cm$^{-3}$ and 1.4 g cm$^{-3}$ (Ng et al., 2007).
For the initially wet AS seeds, all AS particles were completely aqueous and
therefore $f = 1$. For the initially dry AS seeds, before reactions, the AS particles were
completely dry and $f = 0$. After reactions, the AS particles became partially or entirely
deliquesced upon the formation of toluene-derived SOA. The dissolved fraction of AS
particles was regulated by the liquidus curve of the deliquescence relative humidity
(DRH($\varepsilon$)) of AS particles coated with toluene-derived SOA (Smith et al., 2013):
$$f = \begin{cases} \dfrac{\varepsilon(1-\varepsilon_D)}{\varepsilon_D(1-\varepsilon)} & for\ \varepsilon < \varepsilon_D \\ 1 & for\ \ \varepsilon \geq \varepsilon_D \end{cases}$$
(3)

The term $\varepsilon$ is the volume fraction of SOA (Table 1). The term $\varepsilon_D$, representing the
volume fraction of organics at which the mixture of SOA and AS particles deliquesced
at an RH of 68%, was estimated to be 0.75 based on the liquidus curve.
**3. Results and discussion**
**3.1 SOA yields**
Figure 2a shows SOA yields from the photooxidation of toluene on initially wet and
dry AS seed particles as a function of OH exposure. The SOA yield was calculated as
the SOA mass divided by the mass of reacted toluene. The uncertainty in the SOA
yields simply reflected the standard derivation when averaging the SOA mass. In both
cases, SOA yields first exhibited an increase, followed by a decrease as the level of OH
exposure increased. This trend may be due to the transition of functionalization
reactions to fragmentation ones (Kroll et al., 2009; Lambe et al., 2011a). Previous
oxidation flow reactor studies suggest that gas-phase chemistry dominates over
heterogeneous OH oxidation at OH levels below $1.0 \times 10^{12}$ molecules cm$^{-3}$ s (Ortega et
al., 2016; Palm et al., 2016). In this study, the highest OH exposure was $5.28 \times 10^{11}$
molecules cm$^{-3}$ s and heterogeneous oxidation of SOA may not play an important role
in reducing the mass of SOA, although we cannot exclude that it plays a role. In addition,
glyoxal is an important oxidation product of toluene (Kamens et al., 2011). The reactive
uptake of glyoxal has been demonstrated to enhance rather than reduce the SOA mass
(Liggio et al., 2005a). The SOA yields for dry and wet AS seeds were 0.18–0.31 and
0.22–0.36, respectively, significantly higher than the value of 0.0059 observed in an
oxidation flow reactor under comparable conditions (Faust et al., 2017) and the value
of 0.09 obtained in another PAM chamber at 30% RH in the absence of seed particles
(Kang et al., 2007). Faust et al. (2017) attributed their significantly lower yields than
typical literature values of 0.09–0.30 (Lambe et al., 2011a; Ng et al., 2007) to the wall
loss of particles and the fragmentation of organics in their flow reactor. On the other
hand, the SOA yields we obtained are lower than the values of 0.30–0.37 from smog
chamber experiments conducted at a similar temperature, SOA mass loading and OH
exposure but a lower RH with dry AS seeds (Ng et al., 2007; Hildebrandt et al., 2009).
Note that the wall loss of particles was not corrected in this study, so the SOA yields
may be underestimated. As wet and dry AS seeds in this study had similar particle
number size distributions, the wall loss of particles would not affect the comparison of
SOA yield between wet and dry AS seeds.
As shown in Fig. 2a, a higher SOA yield was observed for wet AS seeds than for
dry AS seeds at the same OH exposure and the difference in SOA yield decreased as
the OH exposure increased. The ratio of SOA yields on wet AS seeds to those on dry
AS seeds, the relative SOA yield, was $1.31\pm0.02$ at an OH exposure of $0.47\times10^{11}$
molecules cm$^{-3}$ s but decreased to $1.01\pm0.01$ when the OH exposure was increased to
$5.28\times10^{11}$ molecules cm$^{-3}$ s (Fig. 2b). These ratios are comparable to the $1.19\pm0.05$
observed by Faust et al. (2017) at an OH exposure of approximately $2.0\times10^{11}$ molecules
cm$^{-3}$ s.
The formation of SOA on initially dry AS particles may alter the deliquescence
relative humidity (DRH) of AS particles. Smith et al. (2013) found that when coated
with toluene-derived SOA, the DRH of AS particles decreased from 80% to 58% as the
organic volume fraction increased from 0 to 0.8. Therefore, coating AS particles with
toluene-derived SOA can change the physical state of initially dry AS seeds and
increase the content of $ALW_{AS, dry}$. As shown in Fig. 3a, after reactions, the mass
concentrations of $ALW_{tot}$ (= $ALW_{SOA}$ + $ALW_{AS}$) and $ALW_{SOA}$ increased for both wet
and dry seeds as the OH exposure increased. The uncertainties for $ALW_{SOA}$ and $ALW_{AS}$
were 22% and less than 3%, respectively. They reflect the uncertainties in $\kappa$ and volume
concentrations of AS and SOA. The increase in $ALW_{tot, wet}$ was due to the increase in
$ALW_{SOA, wet}$ while the increase in $ALW_{tot, dry}$ was driven by the increase in $ALW_{AS, dry}$ at
lower OH exposure and by $ALW_{SOA, dry}$ at higher OH exposures. At OH exposure of
$0.47 \times 10^{11}$ molecules $cm^{-3}$ s, $ALW_{AS, dry}$ increased from 0 to 6.2 µg $m^{-3}$ after reactions
due to the partial deliquescence ($f$=0.43) of the originally dry AS particles after SOA
formation. The difference in $ALW_{AS, dry}$ and $ALW_{AS, wet}$ narrowed and the $ALW_{total}$ of
initially dry AS seeds partially resembled those of the wet ones. At OH exposure
between $1.66 \times 10^{11}$ and $5.28 \times 10^{11}$ molecules $cm^{-3}$ s, the total final organic volume
fraction increased to approximately 0.8 and the initially dry AS particles entirely
deliquesced after reactions. Based on the reported SOA yield, initial toluene
concentration, OH exposure and assumed concentrations of AS seeds (~10-40 µg $m^{-3}$)
in Faust et al. (2017), we estimated that an upper limit of 48% of the initially dry AS
seeds has deliquesced in their study. Similar to this study, SOA coatings on seed
particles may change the physical state of initially dry seeds and lower the difference
of SOA yields between initially dry and wet seeds experiments.
The hydrophilic products can partition more readily into initially wet AS seeds than
dry seeds and partially account for the difference in SOA yields. For example, as one
of the important oxidation products, glyoxal was estimated to have an effective Henry's
law constant of $4.52 \times 10^8$ m atm$^{-1}$ for our initially wet AS seeds due to the "salting-in"
effect (Kampf et al., 2013), approximately 3 orders of magnitude higher than that in
pure water (Ip et al., 2009). The uptake rate constant of glyoxal can be calculated as
($\gamma v$A)/4, where $\gamma$ is the uptake coefficient, $v$ is the gas-phase velocity of glyoxal, and A
is the total surface area of AS seeds. The uptake rate constant is $4.5 \times 10^{-4}$ s$^{-1}$ for initially
wet seeds with $\gamma = 2.4 \times 10^{-3}$ estimated from glyoxal uptake in AS seeds at 68% RH
(Liggio et al., 2005b). The average gas-phase glyoxal concentration was modeled to be
4.3 ppb at OH exposure of $0.47 \times 10^{11}$ molecules cm$^{-3}$ s using the Master Chemical
Mechanism v 3.3.1 (Jenkin et al., 2003; Bloss et al., 2005), which would result in
approximately 1.6 µg m$^{-3}$ of glyoxal in particle phase for initially wet AS seeds. If the
particle-phase concentration of glyoxal was assumed to be 0 for initially dry AS seeds,
the enhanced partitioning of glyoxal alone would account for 24.5% of the mass
difference of SOA. Note that other hydrophilic products were not included in this
calculation. This analysis suggests that the enhanced partitioning of hydrophilic
products may play an important role in the difference of SOA yields at low OH
exposures. As discussed above, the initially dry AS seeds approached wet seeds and
reduce the differences between wet and dry SOA yields at high OH exposures.
**3.2 Chemical composition of SOA**
Figure 4 shows the high-resolution mass spectra of SOA for initially wet and dry AS
seeds at OH exposures of $0.47 \times 10^{11}$ molecules $cm^{-3}$ s and $5.28 \times 10^{11}$ molecules $cm^{-3}$ s.
For both types of AS seeds, at an OH exposure of $0.47 \times 10^{11}$ molecules $cm^{-3}$ s, the most
prominent peaks were *m/z* 29 and 43, followed by *m/z* 28 and 44. *m/z* 29 was dominated
by ion $CHO^+$, a tracer for alcohols and aldehydes (Lee et al., 2012). The *m/z* 28 and *m/z*
44 signals, respectively dominated by $CO^+$ and $CO_2^+$, are tracers for organic acids (Ng
et al., 2010). At the OH exposure of $5.28 \times 10^{11}$ molecules $cm^{-3}$ s, the dominant peaks
were *m/z* 28 and 44, followed by *m/z* 29 and 43. The increase of mass fractions of the
oxygen-containing ions in the SOA mass spectra at a relatively high OH exposure
suggests the formation of more oxidized organic aerosols. On the basis of the mass
fraction of ions, Fig. S4 shows that as OH exposure increased, the difference (wet minus
dry) in the spectra of toluene-derived SOA changed from positive in *m/z* 29 ($CHO^+$)
and *m/z* 43 ($C_2H_3O^+$) to *m/z* 28 ($CO^+$) and *m/z* 44 ($CO_2^+$). The increase in OH exposure
resulted in a change from more alcohols or aldehydes to more organic acids in the wet
seeded case when compared to the dry seeded case.

Fragments derived from the AMS data have been extensively used to infer the bulk

compositions and evolution of organic aerosols (Zhang et al., 2005; Ng et al., 2010;
Heald et al., 2010). Here we used the approach of Ng et al. (2010) and plotted the
fractions of the total organic signal at *m/z* 43 ($f_{43}$) vs. *m/z* 44 ($f_{44}$) as well as the triangle
based on the analysis of ambient AMS data (Fig. 5). Ng et al. (2010) proposed that
aging would cause $f_{43}$ and $f_{44}$ to converge toward the triangle apex ($f_{43} = 0.02$, $f_{44} = 0.30$).
For both wet and dry AS seeds, $f_{43}$ first increased and then decreased with the increase

of OH exposure, while $f_{44}$ increased all the time. This reversing trend of $f_{43}$ was the result of the increase and subsequent decrease in $C_2H_3O^+$ (Fig. S5), an indicator of products containing carbonyl functional groups. The $f_{43}$-$f_{44}$ plot supports our earlier assertion that as OH exposure increased, the reaction products changed from earlier-generation products containing carbonyl functional groups dominated to later-generation products containing acidic functional groups dominated. It was also observed for SOA formed from other precursors such as alkanes and naphthalene (Lambe et al., 2011b). Before the decrease in $f_{43}$, SOA formed on wet AS seeds had higher $f_{43}$ and similar $f_{44}$ to SOA formed on dry AS seeds at the same OH exposure. As OH exposure increased, SOA formed on wet AS seeds had higher $f_{44}$ and lower $f_{43}$ than SOA formed on dry AS seeds. In addition, as OH exposure increased, SOA formed on wet AS seeds initially had more earlier-generation products but later had more acidic later-generation products than SOA formed on dry AS seeds, likely due to the enhanced partitioning of these products on initially wet AS seeds and/or enhanced uptake of water-soluble gases through aqueous phase reactions.

Figure 6 shows the changes in H:C and O:C ratios as a function of OH exposure in a Van Krevelen diagram (Heald et al., 2010). The standard deviations for H:C and O:C values, determined for the steady-state periods, were all less than 0.01. The O:C ratios for dry and wet AS seeds were in the ranges of 0.59–0.89 and 0.63–0.95, respectively. At the same OH exposure, SOA on wet AS seeds had both higher O:C ratios and estimated average carbon oxidation state ($OS_C$) ($OS_C \approx 2 \times O:C - H:C$) (Kroll et al., 2011) than dry AS seeds had. Fig. 6 also shows some of the identified SOA products from the

photooxidation of toluene (Bloss et al., 2005; Hamilton et al., 2005; Sato et al., 2007).
The elevated $OS_C$ (exceeding 0.5) could only be due to the formation of highly
oxgenerated small acids such as pyruvic acid ($OS_C = 0.67$), glycolic acid ($OS_C = 1$),
formic acid ($OS_C = 2$), oxalic acid ($OS_C = 3$), malonic acid ($OS_C = 1.33$) and glyoxylic
acid ($OS_C = 2$). Small acids may be important products of toluene-derived SOA at high
OH exposures. Fisseha et al. (2004) found that small organic acids accounted for 20–
45% of SOA from the photooxidation of 1,3,5-trimethylbenzene. The higher $OS_C$ at
high OH exposures for wet AS seeds might suggest that these small acids were more
abundant, likely due to their enhanced retention in the presence of ALW and/or the
more efficient uptake of OH radicals by wet AS seeds and further oxidation reactions
in aqueous phase (Ruehl et al., 2013).
We evaluate whether enhanced uptake of OH radicals on initially wet AS seeds
could explain the difference in oxygen contents, following the method of DeCarlo et al.
(2008). We calculated R, the ratio of the difference in oxygen of OA between the
initially wet and dry AS seed particles to the difference in the total number of OH
collisions with OA at different OH exposures. To obtain R, the uptake coefficient ($\gamma$) of
OH radicals was assumed to be 1 and 0.1/0.8 (lower/upper limit) for initially wet and
dry AS seed particles, respectively (George and Abbatt, 2000). Note that as SOA
formation takes place, the initially dry AS can become wet and the difference in $\gamma$
between initially wet and dry seeds is reduced, especially at higher OH exposures. We
also assumed that each collision of OH with OA resulted in the addition of one oxygen
atom to SOA. A value of R smaller than unity qualitatively indicates that the uptake of
OH radicals can potentially explain the differences in oxygen contents in the dry and
wet experiments. Fig. S6 shows that R is larger than unity at low OH exposures and
smaller than unity at high OH exposures. This analysis suggests that the enhanced OH
uptake may contribute to the difference in oxygen contents between dry and wet cases
at higher OH exposures. At low OH exposures, the enhanced gas-particle partitioning
may dominate the difference.

The change in the slope of H:C vs O:C is consistent with the earlier analysis that

the mechanism of SOA formation changed from functionalization dominated by the
addition of alcohol/peroxide (Heald et al., 2010; Ng et al., 2011) at low exposures to
the addition of both acid and alcohol/peroxide functional groups without fragmentation,
and/or the addition of acid groups with fragmentation at high exposures.
**3.3 Atmospheric implications**
In this work, yields and composition of SOA formed from the photooxidation of toluene
on initially wet and dry AS seeds were compared over a wide range of OH exposures,
covering the transition from functionalization reactions to fragmentation reactions. We
found that the ratio of SOA yield on wet AS seeds to that on dry AS seeds decreased
from 1.31 to 1.01 as the OH exposure increased from $0.47 \times 10^{11}$ to $5.28 \times 10^{11}$ molecules
cm$^{-3}$ s. This decrease coincides with the decrease of differences in ALW between the
wet and dry cases, which may be due to water uptake by SOA as well as the early
deliquescence of dry AS particles as a result of SOA formation. Hence, the SOA yield
and ALW of the initially dry AS seeds approached those of the initially wet AS seeds
as OH exposure and ALW increased.

In addition to relatively higher SOA yields, higher O:C and $OS_c$ of SOA derived

from the photooxidation of toluene were also observed on initially wet AS seeds.
Particularly, the O:C in the presence of initially wet AS seeds could be as high as 0.95.
Chen et al. (2015) observed large gaps between laboratory and ambient measured O:C
of OA and suggested that OA having a high O:C (> 0.6) was required to bridge these
gaps. The multiphase oxidation of toluene in the presence of wet aerosols may be a
pathway to contribute to this gap. However, the relative importance of such chemistry
to the evolution of ambient OA remains unclear.

Our results suggest that dry seeds would quickly turn to at least partially

deliquesced particles upon SOA formation under moderate RH conditions. We only
studied the photooxidation of toluene in the absence of $NO_x$ as it is still a challenge to
study high-NO chemistry in oxidation flow reactors without using atmospherically
irrelevant high concentrations of $NO_x$ (Peng and Jimenez, 2017). However, the ALW
may also be important to SOA formation under high $NO_x$ conditions that preferentially
form highly water-soluble products (Ervens et al., 2011). Since ambient RH is rarely at
such low values that inorganic particles remain dry even after SOA formation, more
laboratory and field studies are needed to elucidate the formation and evolution of OA
under various $NO_x$ conditions at moderate RH.

## Acknowledgments

The work described in this paper was sponsored by the Science Technology and Innovation Committee of Shenzhen Municipality (project no. JCYJ20160401095857424). Zijun Li and ManNin Chan are supported by a Direct Grant for Research (4053159), The Chinese University of Hong Kong and a Research Grants Council grant (RGC 2191111). Chak K. Chan would like to thank the Hong Kong University of Science and Technology for the use of the AMS.

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

**Table 1.** Summary of the results for the initially dry and wet AS seeds experiments.

| OH exposure ($\times 10^{11}$ molecules cm$^{-3}$ s) | [toluene]$_{reacted}$ (ppb) | [toluene]$_{final}$ (ppb) | $\varepsilon$ [a] wet AS | dry AS |
|---|---|---|---|---|
| 0.47 | 32.4 | 106.0 | 0.57 | 0.56 |
| 1.66 | 84.9 | 53.5 | 0.82 | 0.82 |
| 2.97 | 113.1 | 25.3 | 0.83 | 0.85 |
| 4.34 | 126.9 | 11.5 | 0.83 | 0.85 |
| 5.28 | 131.7 | 6.7 | 0.83 | 0.85 |

[a] The volume fraction of organics.

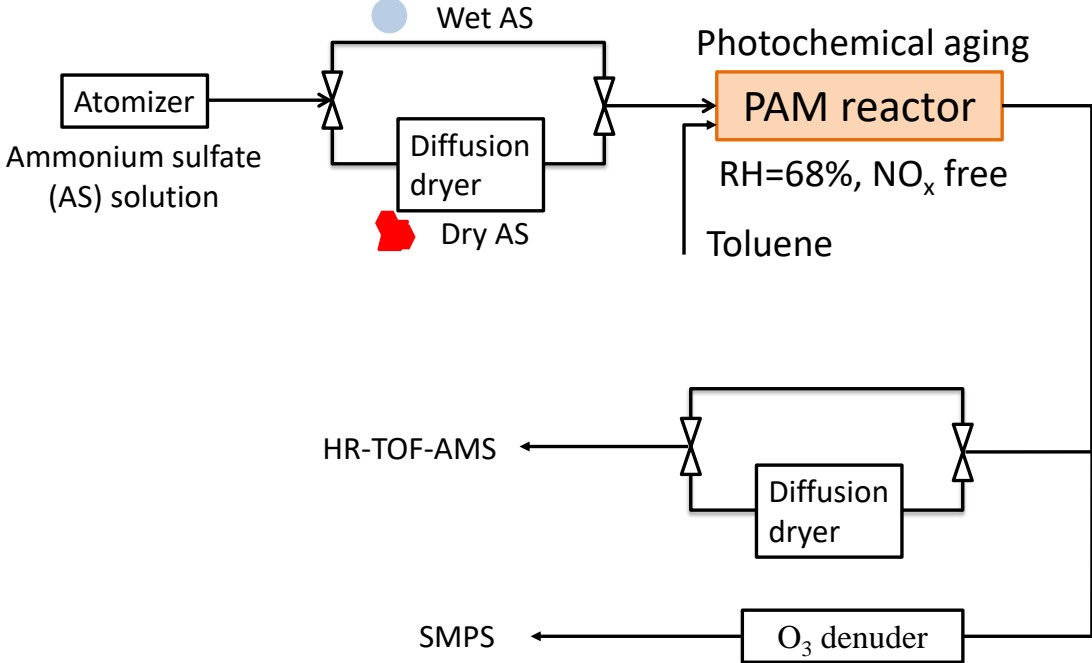


**Fig. 1.** Schematic of the experimental setup. The aqueous ammonium sulfate (AS) seed
particles either passed through a diffusion dryer so that the phase of the seed particles
could be altered or bypassed the diffusion dryer. Either wet or dry AS served as seed
particles for the experiments.

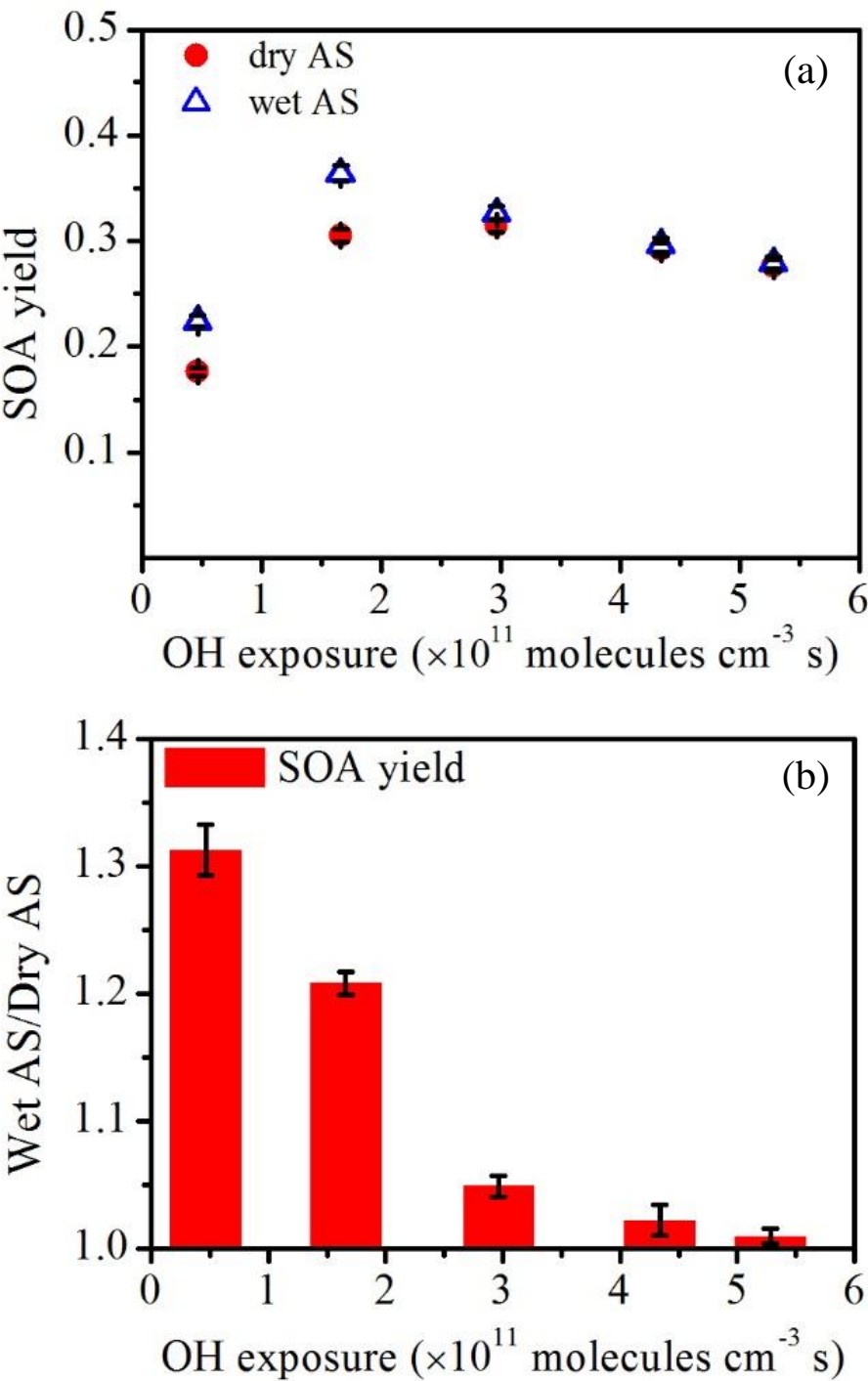


**Fig. 2.** (a) Yield of toluene-derived SOA formed on initially wet and dry AS as a
function of OH exposure. (b) Ratio of SOA yields on initially wet AS to those on
initially dry AS as a function of OH exposure.

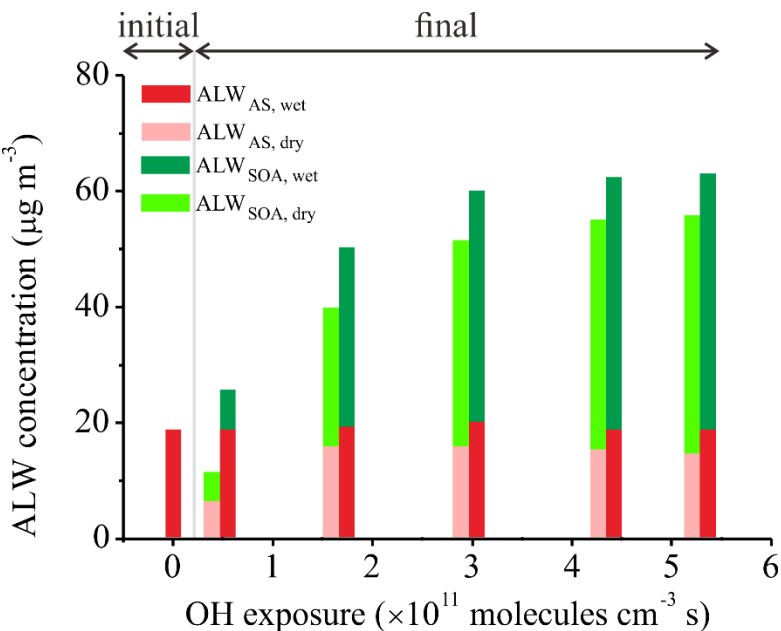


**Fig. 3.** Mass concentration of ALW uptake by AS and toluene-derived SOA before

(initial) and after reactions (final) for both initially wet and dry AS seeds. Adjoining

bars for initially wet and dry seeds have same OH exposures.


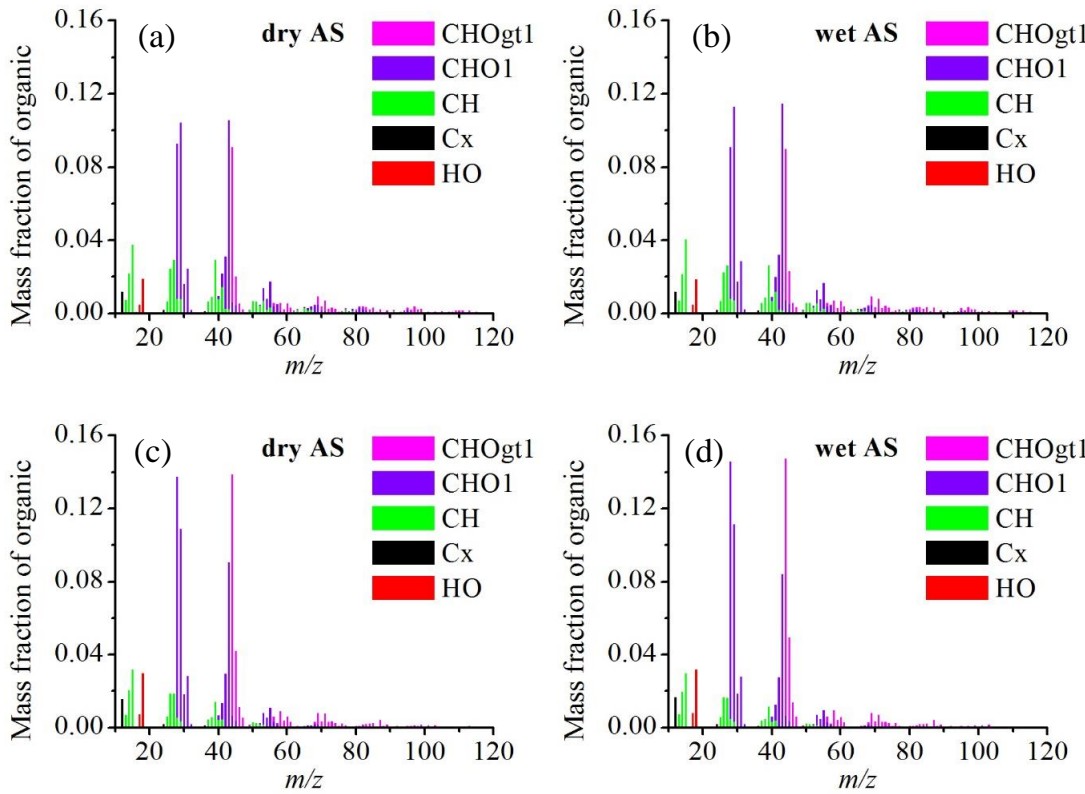


**Fig. 4.** High-resolution mass spectra of toluene-derived SOA on initially wet and dry
AS at an OH exposure of (a, b) $0.47 \times 10^{11}$ molecules cm$^{-3}$ s and (c, d) $5.28 \times 10^{11}$
molecules cm$^{-3}$ s.

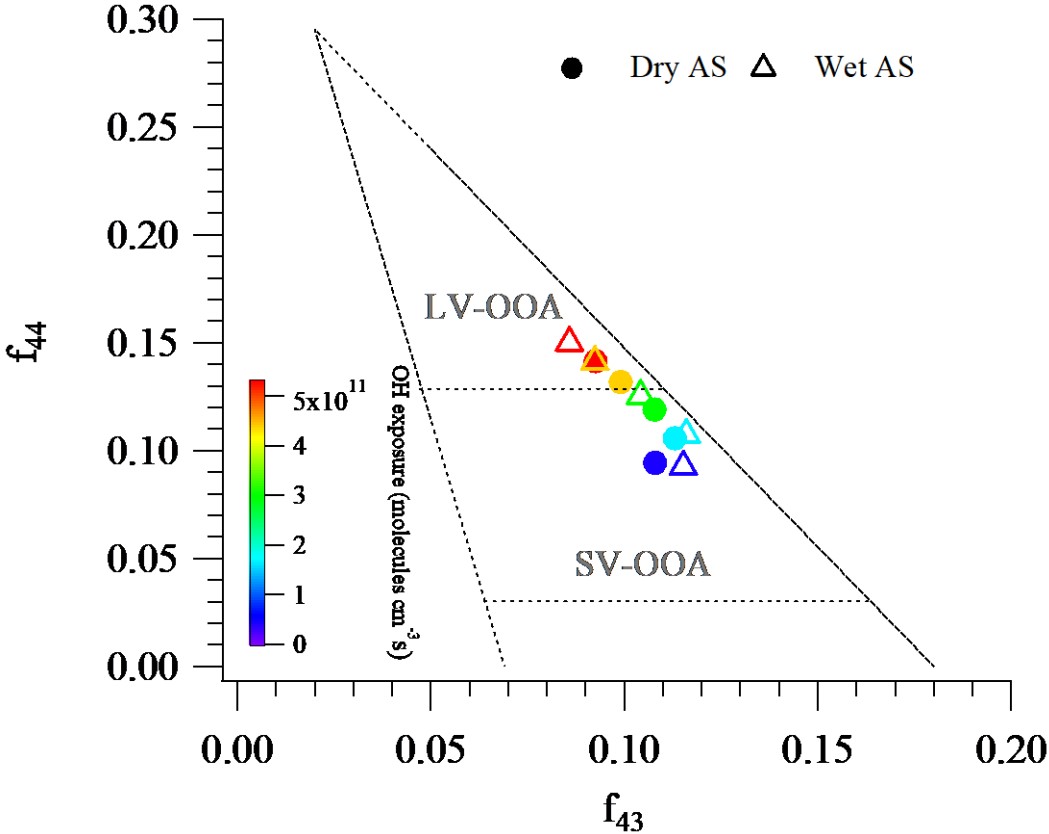


**Fig. 5.** Fractions of total organic signal at *m/z* 43 (f₄₃) vs. *m/z* 44 (f₄₄) from SOA data

obtained in this study together with the triangle plot of Ng et al. (2010). Ambient SV–

OOA and LV–OOA regions are adapted from Ng et al. (2010). Data are colored

according to the OH exposure.


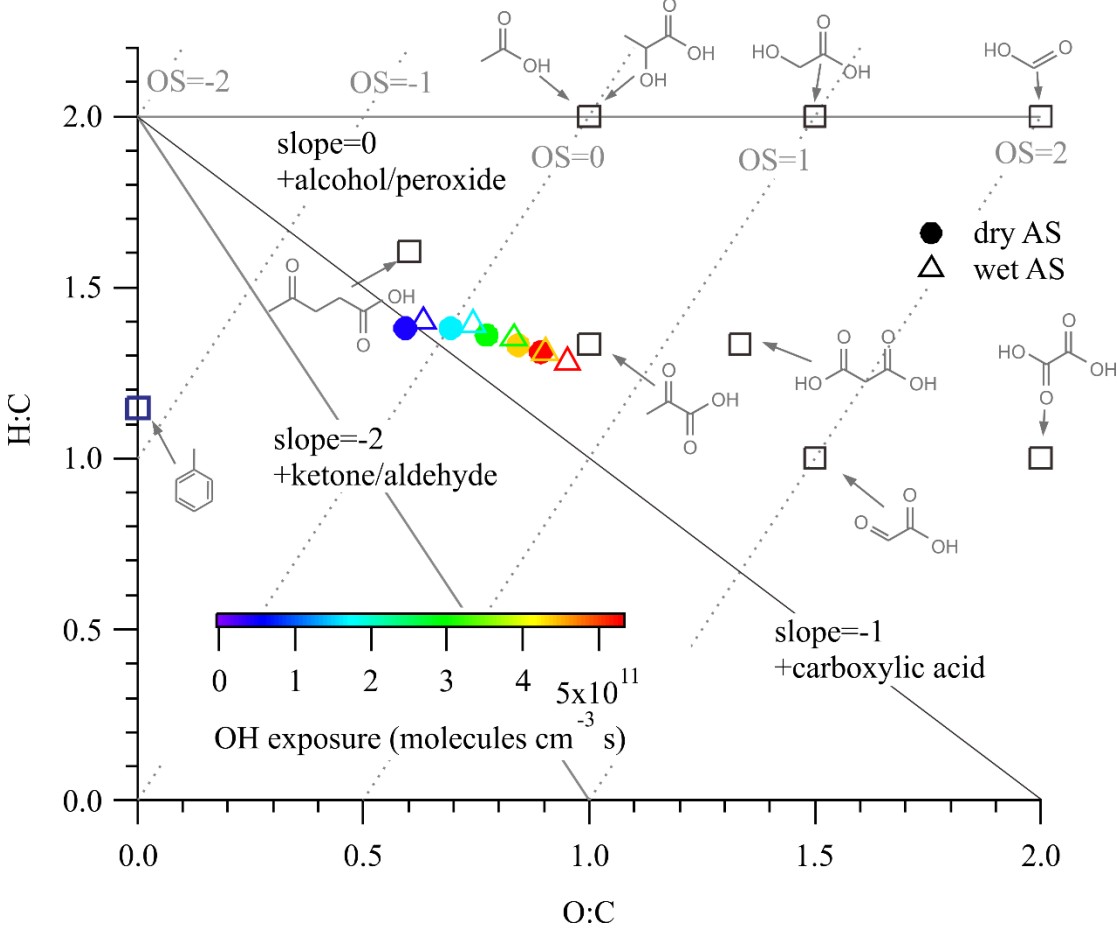


**Fig. 6.** Van Krevelen diagram of SOA derived from the photooxidation of toluene on
initially wet and dry AS seed particles. SOA data are colored according to the OH
exposure. Products identified in toluene-derived SOA are shown in boxes (Bloss et al.,
2005; Hamilton et al., 2005; Sato et al., 2007). Average carbon oxidation states from
Kroll et al. (2011) and functionalization slopes from Heald et al. (2010) are shown for
reference.