# Peer review of "Comparison of secondary organic aerosol formation from"

_Atmospheric Chemistry and Physics, 2017_

## Referee Comment (RC1) · Anonymous Referee #1 · 7 Dec 2017

This manuscript describes laboratory experiments aimed at measuring the yield and composition of SOA formed from the OH oxidation of toluene, in the presence of either initially deliquesced ("wet") or effloresced ("dry") ammonium sulfate (AS) seeds. This work extends the study by Faust et al. (2017), by examining the effects of toluene SOA yield/composition at different OH exposures. The authors certainly highlights an important point that SOA formed onto AS seed particles will lower the deliquescence RH (for initially dry AS) or contribute to additional aerosol liquid water (for initially wet AS). However, my main concern is that the authors only attributed the decrease in relative SOA yield (between dry and wet AS) with increasing OH exposures to the effects of aerosol liquid water, while the experiment design/data presented preclude the isolation

of this effect. In particular, from the composition data presented, this trend could have also arisen due to enhanced contribution of later generation products from the gas-phase oxidation of toluene to the final SOA formed. These later generation products, which the authors mentioned, are generally of smaller molecular weights, would would certainly result in less SOA mass, for the same number of toluene molecules reacted. Additionally, the heterogeneous OH oxidation of the formed SOA is important at higher OH exposures, and could have lead to enhanced loss of SOA mass due to fragmentation for wet AS due to the effects of liquid water on viscosity. A greater discussion is warranted; can other possible explanations for the observed trend in relative SOA yield be ruled out?

Also, the authors only noted that the photo-oxidation of toluene was conducted under low/no NOx conditions in Figure 1 - this is a critical point that needs to be highlighted (especially in the abstract) and warrant a discussion.

Specific comments:

Line 94: The experiment approach is similar (if not identical) to that used in Wong et al. (2015) and Faust et al. (2017) and should be referenced, especially since the introduction of the paper highlights the limitations of these previous studies as motivation of the current work.

Line 114-118: Are the surface area distributions of the "initially" dry and wet AS seed particles? Also, are the total seed particle surface areas significantly high enough that homogeneous nucleation was suppressed?

Line 138: The extent to which the toluene + OH reaction perturbed the OH exposures may be estimated, given that the concentration of toluene added is known.

Line 152: What is the final mixing ratio of toluene in the oxidation flow reactor?

Line 154-155: Were the experiments at different OH exposures conducted separated or in a step-wise manner (e.g. increasing the OH exposures throughout an experiment)?

Line 166-168: Given that fragmentation reactions can lead to the formation of CO2 gas, the filter correction factor should change throughout the experiments.

Line 171-172: It is not clear how the results from Matthews et al. (2008) is comparable to that of the current study.

Line 231-232: It is not clear what is driving the uncertainties in the calculated SOA yields.

Line 246-248: This statement assumes that the wall loss of dry and wet AS particles are identical - is this true?

Lines 257-259: Given that the suppression in the DRH of AS is dependent on the fraction of organics, for the current study, what was the fraction(s) of organics as a function of OH exposure?

Lines 321-322: Do the standard deviations reflect the variability from multiple experiments?

Technical corrections:

Line 81: It is not clear to whom "their" is referring to in this sentence.

---

## Referee Comment (RC2) · Anonymous Referee #2 · 8 Jan 2018

In this manuscript, the authors conducted flow reactor photooxidation experiments of toluene, and examined the SOA formation in the presence of initially wet or dry ammonium sulfate seeds. The approach followed that of Faust et al. (2017) to eliminate the effect of water on the gas-phase oxidation mechanisms, and allows experimenters to focus on the effect of aqueous vs dry seeds on gas-particle partitioning of organics. High resolution aerosol mass spectrometry was used to probe bulk composition, and various mechanisms were proposed to explain the changes in oxidation state, m/z fragment signal fractions and overall mass yields.

The results from this work are qualititatively consistent with those from Faust et al.

[Figure]

There are some questions regarding the oxidation mechanisms that lead to observed changes. My main criticism is that this work report mostly observational results, and lack any detailed mechanistic insights. The results are interesting, and can be explored more in order to provide useful information for both understanding the system and modeling the oxidation. I therefore suggest a few areas to look into, and the manuscript can be strengthened if the following questions are considered.

Major comments:

Oxidation mechanism: The major weakness in the paper is that it largely relies on bulk observation to probe mechanisms. While AMS is useful in obtaining bulk OA information and functional groups, the trends shown here are largely consistent with other aerosol aging observations in the lab or in the field. I am not sure if there are any novel insights in changes of m/z 43 and m/z 44, or the slope of the Van Krevelen diagram. It seems that the wet seed shows slightly different trends, but overall quite insignificant. The authors offer a few potential explanations (e.g. more OH aging, different gas-particle partitioning), but fail to go any deeper. If the explanation is more OH aging, then what is the equivalent change in OH exposure due to a wet seed (e.g. an OH exposure of 1 day at 65% RH is equivalent to an OH exposure of X days at 5%RH). Or, asking the question and framing the results in a different way: What is the increase in OH concentration in the aqueous phase that is required to explain the difference? Is this increase reasonable given the literature on aqueous OH production?

If the enhanced partitioning is due to availability of ALW, one can potentially explain the difference using Henry's law constants. What would the Henry's Law constants of the oxidation products need to be in order to show the difference in SOA yields between wet and dry seeds?

The approach used to calculate ALW separately for AS and toluene assume that they are linearly additive. In a metastable solution of ammonium sulfate, the ionic strength would be very high, and can affect the water solubility of the organics. Would it be

valid to assume then the water uptake that cannot be explained by literature kappa of toluene SOA is associated with AS?

Experimental uncertainties: I am not sure if the experimental uncertainty in yields include only that from quantification of aerosol. The quantification of toluene reacted would play an important role as well, especially since the reported yields (with only uncertainty of SOA) have almost negligible uncertainty. In this work, toluene is not quantified, but the amount reacted is based on estimation of OH exposure. Other work in the literature quantifying yields measure the hydrocarbon precursor directly (using either PTRMS or GC methods). Offline quantification of OH exposure using SO2 and assuming light conditions are constant can be problematic. There needs to be a systematic investigation of the uncertainties, particularly that associated with quantifying hydrocarbon decay.

Minor comments:

Abstract: Should be less focused on specific details of the experiments. I suggest taking some of the numbers out (unless it is a really important number that, for example, a modeler can use or another experimentalist can directly compare). Rather, there may be broader implications other than these results warrant further study. What are the detailed mechanistic insights? What further developments are needed to better understand water uptake? Just a few sentences would suffice.

Line 36: m/z 29, 43, 44 are specific to the aerosol mass spectrometer (AMS).

Line 47-48: these references might not quite reflect the current state of knowledge. For example, Tsigaridis et al. (ACP, 2014) did a model intercomparison and found that the secondary nature of OA can be reproduced, but not the overall amount. Shrivastava et al (ACP, 2011) shows that the total SOA amount can be reproduced, but require some model tuning. Overall, there may be too many studies to cite for a simple argument here, but the de Gouw et al. and Volkamer et al. studies are quite out of date at this point.

Line 60: I would also add that understanding water uptake of SOA is important for estimating its loss by wet deposition, which is highly unknown at this point.

Line 97: is the silica gel diffusion dryer manufactured by TSI? If so, consider including the model number. Is the 30% outlet RH experimentally verified? I would imagine the outlet RH would be a function of the inlet RH

Line 103: Is it correct that the ALW is estimated using the method described later? If so, please mention.

Lines 140-146: Based on the OH exposure calculated, what is the amount of toluene consumed?

Line 166-167: is this filtered air flow with or without toluene and/or oxidation products?

Line 177: Just to make sure that sulfate is indeed coated with SOA, it would be great to show size distributions: Either a unimodal SMPS number size distribution showing no nucleation, or the PTOF on the AMS showing that organics and sulfate are in the same mode.

Line 187: Is it really evaporation of the organics? Would 8% mass loss due to evaporation be enough to show up in shifts in particle mode or median diameter (it would be around a ∼2% change in diameter)?

Line 252: I suggest keeping all OH exposure numbers to 10ˆ11 molec cm-3 s. For example, I recommend changing 4.66 x 10ˆ10 to 0.47 x 10ˆ11.

Line 290 – 292: The authors should be cautious about observations of oligomers on the AMS. The relatively low signal above m/z 80 does not mean there is little to no oligomerization. The AMS vaporizer at 600 C causes extensive thermal decomposition.

Line 298-303: Would a plot f(28+44) vs f(29+43) look different from f44 vs f43? The conversion of alcohols and aldehydes to acids described in the previous paragraph is fundamentally driving the trend of f44 and f43. So these two paragraphs are both

conveying a similar observation, which is expected given the extensive literature on AMS description of aerosol aging. So I suggest condensing the discussion.

Line 314-315, and Fig. S4: The differences seem very minute to me and may not be statistically significant. The y-scale is misleading, and should start at zero.

Line 314-315: If the trend is due to enhanced partitioning of water soluble organics to ALW, wouldn't the

Line 327: It looks like the subscript "C" in OSC is not capitalized, where it has been capitalized in other instances in the manuscript.

References: Tsigaridis, K., Daskalakis, N., Kanakidou, M., Adams, P. J., Artaxo, P., Bahadur, R., Balkanski, Y., Bauer, S. E., Bellouin, N., Benedetti, A., Bergman, T., Berntsen, T. K., Beukes, J. P., Bian, H., Carslaw, K. S., Chin, M., Curci, G., Diehl, T., Easter, R. C., Ghan, S. J., Gong, S. L., Hodzic, A., Hoyle, C. R., Iversen, T., Jathar, S., Jimenez, J. L., Kaiser, J. W., Kirkevåg, A., Koch, D., Kokkola, H., Lee, Y. H., Lin, G., Liu, X., Luo, G., Ma, X., Mann, G. W., Mihalopoulos, N., Morcrette, J.-J., Müller, J.-F., Myhre, G., Myriokefalitakis, S., Ng, N. L., O'Donnell, D., Penner, J. E., Pozzoli, L., Pringle, K. J., Russell, L. M., Schulz, M., Sciare, J., Seland, Ø., Shindell, D. T., Sillman, S., Skeie, R. B., Spracklen, D., Stavrakou, T., Steenrod, S. D., Takemura, T., Tiitta, P., Tilmes, S., Tost, H., van Noije, T., van Zyl, P. G., von Salzen, K., Yu, F., Wang, Z., Wang, Z., Zaveri, R. A., Zhang, H., Zhang, K., Zhang, Q., and Zhang, X.: The AeroCom evaluation and intercomparison of organic aerosol in global models, Atmos. Chem. Phys., 14, 10845-10895, https://doi.org/10.5194/acp-14-10845-2014, 2014.

Shrivastava, M., Fast, J., Easter, R., Gustafson Jr., W. I., Zaveri, R. A., Jimenez, J. L., Saide, P., and Hodzic, A.: Modeling organic aerosols in a megacity: comparison of simple and complex representations of the volatility basis set approach, Atmos. Chem. Phys., 11, 6639-6662, https://doi.org/10.5194/acp-11-6639-2011, 2011.

---

## Author Comment (AC1) · 2 Mar 2018

Response to Reviewer #1

**General comments:**

**Q1:** This manuscript describes laboratory experiments aimed at measuring the yield and composition of SOA formed from the OH oxidation of toluene, in the presence of either initially deliquesced ("wet") or effloresced ("dry") ammonium sulfate (AS) seeds. This work extends the study by Faust et al. (2017), by examining the effects of toluene SOA yield/composition at different OH exposures. The authors certainly highlights an important point that SOA formed onto AS seed particles will lower the deliquescence RH (for initially dry AS) or contribute to additional aerosol liquid water (for initially wet AS). However, my main concern is that the authors only attributed the decrease in relative SOA yield (between dry and wet AS) with increasing OH exposures to the effects of aerosol liquid water, while the experiment design/data presented preclude the isolation of this effect. In particular, from the composition data presented, this trend could have also arisen due to enhanced contribution of later generation products from the gas-phase oxidation of toluene to the final SOA formed. These later generation products, which the authors mentioned, are generally of smaller molecular weights, which would certainly result in less SOA mass, for the same number of toluene molecules reacted. Additionally, the heterogeneous OH oxidation of the formed SOA is important at higher OH exposures, and could have led to enhanced loss of SOA mass due to fragmentation for wet AS due to the effects of liquid water on viscosity. A greater discussion is warranted; can other possible explanations for the observed trend in relative SOA yield be ruled out?

R1: We emphasize that at moderate and atmospheric relevant RH, aerosol liquid water will exist and play an important role and it cannot be avoided, even if the experiment started with dry AS. We would argue that the hygroscopic properties of AS and SOA naturally leads to the PRESENCE of water in particles under such conditions. It is not feasible to isolate the role of water from SOA formation under these moderate RHs. We attempted to estimate the amount of ALW at different OH exposures based on the sulfate and OA data obtained from the AMS.

Yes, the later generation products would result in less absolute SOA yields, which is confirmed by the observed decrease in the absolute SOA yields with the increase of OH exposure for both dry and wet experiments (Fig. 2).

In general, decrease in SOA yield can be attributed to fragmentation in gas phase and heterogeneous reactions. Previous oxidation flow reactor studies investigating the aging of ambient air in urban and forest areas suggest that gas-phase chemistry dominates over heterogeneous OH oxidation at OH levels below $1.0\times10^{12}$ molecules $cm^{-3}$ s (Ortega et al., 2016; Palm et al., 2016). In this study, the highest OH exposure was $5.28\times10^{11}$ molecules $cm^{-3}$ s and heterogeneous oxidation of SOA may not play an important role in reducing the mass of SOA. In addition, glyoxal is an important oxidation product of toluene (Kamens et al., 2011). The reactive uptake of glyoxal has been demonstrated to enhance rather than reduce the SOA mass (Liggio et al., 2005). The following sentences have been added to the revised manuscript.

"Previous oxidation flow reactor studies suggest that gas-phase chemistry dominates over heterogeneous OH oxidation at OH levels below $1.0\times10^{12}$ molecules $cm^{-3}$ s (Ortega et al., 2016; Palm et al., 2016). In this study, the highest OH exposure was $5.28\times10^{11}$ molecules $cm^{-3}$ s and heterogeneous oxidation of SOA may not play an important role in reducing the mass of SOA, although we cannot exclude that it plays a role. In addition, glyoxal is an important oxidation product of toluene (Kamens et al., 2011). The reactive uptake of glyoxal has been demonstrated to enhance rather than reduce the SOA mass (Liggio et al., 2005a)." (Line 264-271).

**Q2:** Also, the authors only noted that the photo-oxidation of toluene was conducted under low/no NOx conditions in Figure 1 - this is a critical point that needs to be highlighted (especially in the abstract) and warrant a discussion.

R2: In a PAM, OH dominated reactions even at high $NO_x$. NO at ambient high levels is rapidly oxidized by the high concentrations of OH, $HO_2$ and $O_3$ and hence the reaction would still be OH dominant. To study $NO_x$ chemistry, extreme unrealistically high concentration of $NO_x$ (e.g. a few ppm) is used, which would render the reactions atmospherically irrelevant. Hence, we only studied the photooxidation of toluene in the absence of $NO_x$ as it is still a challenge to study $NO_x$ reactions in oxidation flow reactors

without using atmospherically irrelevant high concentrations of NOx (Peng and Jimenez, 2017). However, aerosol liquid water may also be important to SOA formation under high NOx conditions that preferentially form highly water-soluble products (Ervens et al., 2011). Further studies are needed to elucidate the interplay between SOA and ALW under high $NO_x$ conditions. The following text has been added for clarification.

"in the absence of $NO_x$" (Line 25-26; Line 137).

"We only studied the photooxidation of toluene in the absence of $NO_x$ as it is still a challenge to study high-NO chemistry in oxidation flow reactors without using atmospherically irrelevant high concentrations of $NO_x$ (Peng and Jimenez, 2017). However, the ALW may also be important to SOA formation under high $NO_x$ conditions that preferentially form highly water-soluble products (Ervens et al., 2011)." (Line 440-445).

"under various $NO_x$ conditions at moderate RH" (Line 447-448).

**Specific comments:**

**Q3:** Line 94: The experiment approach is similar (if not identical) to that used in Wong et al. (2015) and Faust et al. (2017) and should be referenced, especially since the introduction of the paper highlights the limitations of these previous studies as motivation of the current work.

R3: The following sentence has been added to the revised manuscript.

"similar to that used in Wong et al. (2015) and Faust et al. (2017)" (Line 109-110).

**Q4:** Line 114-118: Are the surface area distributions of the "initially" dry and wet AS seed particles? Also, are the total seed particle surface areas significantly high enough that homogeneous nucleation was suppressed?

R4: Yes. These are distributions of initially dry and wet AS seed particles. The term "initially" was added to the revised manuscript for clarification.

The total seed particle surface areas are high enough to suppress nucleation. As shown in the following figure (now Fig. S2), at the OH exposure of $4.66\times10^{10}$ molecules $cm^{-3}$ s, the particle number distributions for both cases are unimodal,

indicating no nucleation. The following sentence was added to the revised manuscript.

"The unimodal size distributions of particle numbers show the SOA formation on AS seed particles without much nucleation mode particles (Fig. S2)." (Line 198-200)

[Figure]

**Q5:** Line 138: The extent to which the toluene + OH reaction perturbed the OH exposures may be estimated, given that the concentration of toluene added is known.

R5: The reduction in OH exposure due to the toluene + OH reaction was estimated to range from 15% at the highest OH exposure to 25% at the lowest OH exposure, using the method of Peng et al. (2016). We assume that this reduction is the same for dry and wet seeds and will not influence the relative SOA yields.

The sentence "*The addition of toluene may reduce the OH exposure.*" has been revised and now reads:

"The reduction in OH exposure due to the addition of toluene was estimated to range from 15% at the highest OH exposure to 25% at the lowest OH exposure, using the method of Peng et al. (2016)." (Line 157-160).

**Q6:** Line 152: What is the final mixing ratio of toluene in the oxidation flow reactor?

R6: The calculated final mixing ratio of toluene is now provided in Table 1. The following sentence has been added:

"The reacted and final concentrations of toluene were calculated from the OH exposure and the rate constant of the reaction between toluene and OH (Atkinson and Arey, 2003) (Table 1)." (Line 172-174).

**Table 1.** Summary of the results for the initially dry and wet AS seeds experiments.

| OH exposure ($\times 10^{11}$ molecules cm$^{-3}$ s) | [toluene]$_{reacted}$ (ppb) | [toluene]$_{final}$ (ppb) | $\varepsilon$ [a] | |
|---|---|---|---|---|
| | | | wet AS | dry AS |

| | | | | |
|---|---|---|---|---|
| 0.47 | 32.4 | 106.0 | 0.57 | 0.56 |
| 1.66 | 84.9 | 53.5 | 0.82 | 0.82 |
| 2.97 | 113.1 | 25.3 | 0.83 | 0.85 |
| 4.34 | 126.9 | 11.5 | 0.83 | 0.85 |
| 5.28 | 131.7 | 6.7 | 0.83 | 0.85 |

[a] The volume fraction of organics.

**Q7:** Line 154-155: Were the experiments at different OH exposures conducted separated or in a step-wise manner (e.g. increasing the OH exposures throughout an experiment)?

R7: The experiments were conducted with a step-wise increase in OH exposure. "*at each of*" now reads "with a step-wise increase in". (Line 178).

**Q8:** Line 166-168: Given that fragmentation reactions can lead to the formation of $CO_2$ gas, the filter correction factor should change throughout the experiments.

R8: The concentration of formed $CO_2$ would be less than 1 ppm even all the toluene (~138 ppb) was oxidized to $CO_2$. An 1 ppm increase of $CO_2$ can only lead to 0.0006 ug/m$^3$ increase of SOA and has no detectable influence on O:C ratios. Therefore, the influence of the formation of this extremely low concentration of $CO_2$ on AMS data analysis is negligible.

**Q9:** Line 171-172: It is not clear how the results from Matthews et al. (2008) is comparable to that of the current study.

R9: Similar to Matthews et al. (2008), AS seed particles were also coated by liquid state of SOA.

"*A CE of 1 was used for processing all AMS data since the concentration of sulfate measured with the AMS varied by less than 5% of the average mass of sulfate after coated by SOA for both wet and dry AS seeds conditions.*" now reads:

"The toluene-derived SOA in these experiments was therefore liquid-like. The unimodal size distributions of particle numbers show the SOA formation on AS seed particles without much nucleation mode particles (Fig. S2). A CE of 1 was used for processing all AMS data since the AS seed particles were coated by liquid SOA. The adoption of this CE value was supported by that the concentration of sulfate measured with the AMS varied by less than 5% of the average mass of sulfate after coated by

SOA for both wet and dry AS seeds conditions." (Line 197-204).

**Q10:** Line 231-232: It is not clear what is driving the uncertainties in the calculated SOA yields.

R10: The reported uncertainties were solely due to the standard derivations when averaging the concentrations of SOA.

  "*The uncertainty in the SOA yields fully reflected the uncertainty in the calculation of the SOA mass*" now reads: "The uncertainty in the SOA yields simply reflected the standard derivation when averaging the SOA mass" (Line 260).

**Q11:** Line 246-248: This statement assumes that the wall loss of dry and wet AS particles are identical - is this true?

R11: According to McMurry and Grosjean (1985), the wall loss coefficient was size dependent. We assumed the wall losses of wet and dry particles are similar considering their similar size distributions of particle number.

**Q12:** Lines 257-259:   Given that the suppression in the DRH of AS is dependent on the fraction of organics, for the current study, what was the fraction(s) of organics as a function of OH exposure?

R12: The volume fractions of organics are now provided in Table 1.

"(Table 1)" was added to the revised manuscript. (Line 250).

**Q13:** Lines 321-322: Do the standard deviations reflect the variability from multiple experiments?

R13: They reflected the variability of the steady-state periods.

 "determined for the steady-state periods" was added to the revised manuscript. (Line 382).

**Technical comments:**

**Q14**: Line 81: It is not clear to whom "their" is referring to in this sentence.

R14: "their study" was changed to "Faust et al. (2017)". (Line 95).

References:

Ervens, B., Turpin, B. J., and Weber, R. J.: Secondary organic aerosol formation in

cloud droplets and aqueous particles (aqSOA): a review of laboratory, field and model studies, Atmos. Chem. Phys., 11, 11069-11102, 10.5194/acp-11-11069-2011, 2011.

Kamens, R. M., Zhang, H. F., Chen, E. H., Zhou, Y., Parikh, H. M., Wilson, R. L., Galloway, K. E., and Rosen, E. P.: Secondary organic aerosol formation from toluene in an atmospheric hydrocarbon mixture: Water and particle seed effects, Atmos Environ, 45, 2324-2334, DOI 10.1016/j.atmosenv.2010.11.007, 2011.

Liggio, J., Li, S.-M., and McLaren, R.: Heterogeneous Reactions of Glyoxal on Particulate Matter: Identification of Acetals and Sulfate Esters, Environ Sci Technol, 39, 1532-1541, 10.1021/es048375y, 2005.

McMurry, P. H., and Grosjean, D.: Gas and aerosol wall losses in Teflon film smog chambers, Environ Sci Technol, 19, 1176-1182, 10.1021/es00142a006, 1985.

Ortega, A. M., Hayes, P. L., Peng, Z., Palm, B. B., Hu, W., Day, D. A., Li, R., Cubison, M. J., Brune, W. H., Graus, M., Warneke, C., Gilman, J. B., Kuster, W. C., de Gouw, J., Gutiérrez-Montes, C., and Jimenez, J. L.: Real-time measurements of secondary organic aerosol formation and aging from ambient air in an oxidation flow reactor in the Los Angeles area, Atmos. Chem. Phys., 16, 7411-7433, 10.5194/acp-16-7411-2016, 2016.

Palm, B. B., Campuzano-Jost, P., Ortega, A. M., Day, D. A., Kaser, L., Jud, W., Karl, T., Hansel, A., Hunter, J. F., Cross, E. S., Kroll, J. H., Peng, Z., Brune, W. H., and Jimenez, J. L.: In situ secondary organic aerosol formation from ambient pine forest air using an oxidation flow reactor, Atmos. Chem. Phys., 16, 2943-2970, 10.5194/acp-16-2943-2016, 2016.

Peng, Z., Day, D. A., Ortega, A. M., Palm, B. B., Hu, W., Stark, H., Li, R., Tsigaridis, K., Brune, W. H., and Jimenez, J. L.: Non-OH chemistry in oxidation flow reactors for the study of atmospheric chemistry systematically examined by modeling, Atmos. Chem. Phys., 16, 4283-4305, 10.5194/acp-16-4283-2016, 2016.

Peng, Z., and Jimenez, J. L.: Modeling of the chemistry in oxidation flow reactors with high initial NO, Atmos. Chem. Phys., 17, 11991-12010, 10.5194/acp-17-11991-2017, 2017.

---

## Author Comment (AC2) · 2 Mar 2018

Response to Reviewer #2

**General comments:**

In this manuscript, the authors conducted flow reactor photooxidation experiments of toluene, and examined the SOA formation in the presence of initially wet or dry ammonium sulfate seeds. The approach followed that of Faust et al. (2017) to eliminate the effect of water on the gas-phase oxidation mechanisms, and allows experimenters to focus on the effect of aqueous vs dry seeds on gas-particle partitioning of organics. High resolution aerosol mass spectrometry was used to probe bulk composition, and various mechanisms were proposed to explain the changes in oxidation state, m/z fragment signal fractions and overall mass yields.

The results from this work are qualititatively consistent with those from Faust et al.

There are some questions regarding the oxidation mechanisms that lead to observed changes. My main criticism is that this work report mostly observational results, and lack any detailed mechanistic insights. The results are interesting, and can be explored more in order to provide useful information for both understanding the system and modeling the oxidation. I therefore suggest a few areas to look into, and the manuscript can be strengthened if the following questions are considered.

R: Many thanks for the suggestions. We emphasize that at moderate and atmospherically relevant RH, aerosol liquid water will exist and play an important role and it cannot be avoided, even for experiments started with dry AS seeds. The hygroscopic properties of AS and SOA naturally lead to the PRESENCE of water in particles in both dry and wet experiments. Our goal is not to provide mechanistic details of the reactions. Nevertheless, we appreciate the comments and suggestions of the reviewer to improve the manuscript.

**Major comments:**

**Q1:** Oxidation mechanism: The major weakness in the paper is that it largely relies on bulk observation to probe mechanisms. While AMS is useful in obtaining bulk OA information and functional groups, the trends shown here are largely consistent with other aerosol aging observations in the lab or in the field.   I am not sure if there are

any novel insights in changes of m/z 43 and m/z 44, or the slope of the Van Krevelen diagram. It seems that the wet seed shows slightly different trends, but overall quite insignificant. The authors offer a few potential explanations (e.g. more OH aging, different gas-particle partitioning), but fail to go any deeper. If the explanation is more OH aging, then what is the equivalent change in OH exposure due to a wet seed (e.g. an OH exposure of 1 day at 65% RH is equivalent to an OH exposure of X days at 5%RH). Or, asking the question and framing the results in a different way:    What is the increase in OH concentration in the aqueous phase that is required to explain the difference? Is this increase reasonable given the literature on aqueous OH production?

R1: AMS can provide insights to the overall evolution of OA and show the difference in bulk composition between initially dry and wet seeds without molecular level identification. The difference in SOA mass and composition between experiments with initially dry and wet seeds may be due to the enhanced gas-particle partitioning and/or enhanced OH aging in heterogeneous reactions. Since our experiments were conducted at 68%RH but not at 5%RH in both dry and wet cases, we cannot compare the results at different RH as proposed by the reviewer. Furthermore, though we cannot directly measure or calculate the OH concentration in the aqueous phase, we estimate the uptake of OH radicals to indirectly reflect the effects of enhanced OH aging on oxygen contents.

Specifically, we evaluate whether the enhanced uptake of OH radicals on initially wet AS seeds could explain the difference in oxygen contents, following the method of DeCarlo et al. (2008). We calculated R, the ratio of the difference in oxygen of OA between the initially wet and dry AS seed particles to the difference in the total number of OH collisions with OA at different OH exposures. To obtain R, the uptake coefficient ($\gamma$) of OH radicals was assumed to be 1 and 0.1/0.8 (lower/upper limit) for initially wet and dry AS seed particles, respectively (George and Abbatt, 2000). Note that as SOA formation takes place, the initially dry AS can become wet and the difference in $\gamma$ between initially wet and dry seeds is reduced, especially at higher OH exposures. Without molecular level information on the organics, we assumed that each collision of OH with OA resulted in the addition of one oxygen atom to SOA. A value of R smaller than unity qualitatively indicates that the uptake of OH radicals can potentially explain

the differences in oxygen contents in the dry and wet experiments.

The following figure (now Figure S6) shows that R is larger than unity at low OH exposures and smaller than unity at high OH exposures. This analysis suggests that the enhanced OH uptake may contribute to the difference in oxygen contents between dry and wet cases at higher OH exposures. At low OH exposures, the enhanced gas-particle partitioning may dominate the difference.

[Figure]

The following sentences have been added to the revised manuscript.

"We evaluate whether enhanced uptake of OH radicals on initially wet AS seeds could explain the difference in oxygen contents, following the method of DeCarlo et al. (2008). We calculated R, the ratio of the difference in oxygen of OA between the initially wet and dry AS seed particles to the difference in the total number of OH collisions with OA at different OH exposures. To obtain R, the uptake coefficient ($\gamma$) of OH radicals was assumed to be 1 and 0.1/0.8 (lower/upper limit) for initially wet and dry AS seed particles, respectively (George and Abbatt, 2000). Note that as SOA formation takes place, the initially dry AS can become wet and the difference in $\gamma$ between initially wet and dry seeds is reduced, especially at higher OH exposures. We

also assumed that each collision of OH with OA resulted in the addition of one oxygen atom to SOA. A value of R smaller than unity qualitatively indicates that the uptake of OH radicals can potentially explain the differences in oxygen contents in the dry and wet experiments. Fig. S6 shows that R is larger than unity at low OH exposures and smaller than unity at high OH exposures. This analysis suggests that the enhanced OH uptake may contribute to the difference in oxygen contents between dry and wet cases at higher OH exposures. At low OH exposures, the enhanced gas-particle partitioning may dominate the difference." (Line 398-414).

**Q2:** If the enhanced partitioning is due to availability of ALW, one can potentially explain the difference using Henry's law constants. What would the Henry's Law constants of the oxidation products need to be in order to show the difference in SOA yields between wet and dry seeds?

R2: The hydrophilic products should partition more readily into initially wet AS seeds than dry seeds and partially account for the difference in SOA yields. Both ALW and the Henry's law constant are relevant. Instead of focusing on the Henry's law constant alone, we focus on the uptake of glyoxal, a gas phase oxidation product of toluene oxidation, to illustrate the effects of enhanced partitioning of oxidation products on SOA yields. The following text has been added to the revised manuscript to estimate the effect of enhanced partitioning on SOA yields.

"The hydrophilic products should partition more readily into initially wet AS seeds than dry seeds and partially account for the difference in SOA yields. For example, as one of the important oxidation products, glyoxal was estimated to have an effective Henry's law constant of $4.52\times10^8$ m atm$^{-1}$ for our initially wet AS seeds due to the "salting-in" effect (Kampf et al., 2013), approximately 3 orders of magnitude higher than that in pure water (Ip et al., 2009). The uptake rate constant of glyoxal can be calculated as $(\gamma v A)/4$, where $\gamma$ is the uptake coefficient, $v$ is the gas-phase velocity of glyoxal, and A is the total surface area of AS seeds. The uptake rate constant is $4.5\times10^{-4}$ s$^{-1}$ for initially wet seeds with $\gamma = 2.4\times10^{-3}$ estimated from glyoxal uptake in AS seeds at 68% RH (Liggio et al., 2005b). The average gas-phase glyoxal concentration was modeled to be 4.3 ppb at OH exposure of $0.47\times10^{11}$ molecules cm$^{-3}$ s using the Master

Chemical Mechanism v 3.3.1 (Jenkin et al., 2003; Bloss et al., 2005), which would result in approximately 1.6 µg m$^{-3}$ of glyoxal in particle phase for initially wet AS seeds. If the particle-phase concentration of glyoxal was assumed to be 0 for initially dry AS seeds, the enhanced partitioning of glyoxal alone would account for 24.5% of the mass difference of SOA. Note that other hydrophilic products were not included in this calculation. This analysis suggests that the enhanced partitioning of hydrophilic products may play an important role in the difference of SOA yields at low OH exposures. As discussed above, the initially dry AS seeds approached wet seeds and reduce the differences between wet and dry SOA yields at high OH exposures. " (Line 317-336).

**Q3:** The approach used to calculate ALW separately for AS and toluene assume that they are linearly additive. In a metastable solution of ammonium sulfate, the ionic strength would be very high, and can affect the water solubility of the organics. Would it be valid to assume then the water uptake that cannot be explained by literature kappa of toluene SOA is associated with AS?

R3: The approach used in this study has been found to be adequate to well estimate the hygroscopic growth of inorganic and organic mixtures in laboratory and ambient studies, even at relatively low RH (Choi and Chan, 2002; Cheung et al., 2015; Svenningsson et al., 2006; Nguyen et al., 2016).

**Q4:** Experimental uncertainties: I am not sure if the experimental uncertainty in yields include only that from quantification of aerosol. The quantification of toluene reacted would play an important role as well, especially since the reported yields (with only uncertainty of SOA) have almost negligible uncertainty. In this work, toluene is not quantified, but the amount reacted is based on estimation of OH exposure. Other work in the literature quantifying yields measure the hydrocarbon precursor directly (using either PTRMS or GC methods). Offline quantification of OH exposure using SO2 and assuming light conditions are constant can be problematic. There needs to be a systematic investigation of the uncertainties, particularly that associated with quantifying hydrocarbon decay.

R4: The experimental setup follows Wong et al. (2015) and Faust et al. (2017). The

flow conditions were exactly the same for initially wet and dry seeds. The calibration and toluene photooxidation experiments were conducted over three days. The light condition was not expected to change in such a short period. Furthermore, we focus on the relative SOA yields, which are not expected to be much affected by the uncertainties in toluene quantification since the initial concentrations of toluene and OH exposures were the same for both cases. In offline calibration of OH exposures, the addition of toluene would perturb the calculated OH exposures. The reduction in OH exposure due to the toluene + OH reaction was estimated to range from 15% at the highest OH exposure to 25% at the lowest OH exposure, using the method of Peng et al. (2016). Nevertheless, we assume that this reduction is the same for dry and wet seeds and will not influence the relative SOA yields.

For clarification, the sentence "*The addition of toluene may reduce the OH exposure.*" has been revised and now reads:

"The reduction in OH exposure due to the addition of toluene was estimated to range from 15% at the highest OH exposure to 25% at the lowest OH exposure, using the method of Peng et al. (2016)." (Line 157-160).

The following text was also added for clarification.

"The flow and light conditions were the same for initially wet and dry seeds. Therefore, the quantification of toluene would not introduce uncertainties to the relative SOA yields described in Section 3.1 as the initial concentrations of toluene and OH exposures were the same for both cases." (Line 174-177).

**Minor comments:**

**Q5:** Abstract:    Should be less focused on specific details of the experiments. I suggest taking some of the numbers out (unless it is a really important number that, for example, a modeler can use or another experimentalist can directly compare). Rather, there may be broader implications other than these results warrant further study. What are the detailed mechanistic insights? What further developments are needed to better understand water uptake? Just a few sentences would suffice.

R5: Some parts of the abstract have been rewritten as suggested.

*"At an OH exposure of 4.66×10^10 molecules cm^-3 s, the ratio of the SOA yield on wet AS seeds to that on dry AS seeds was 1.31±0.02. However, this ratio decreased to 1.01±0.01 at an OH exposure of 5.28×10^11 molecules cm^-3 s. The decrease in the ratio of SOA yields as the increase of OH exposure may be due to the early deliquescence of initially dry AS seeds after coated by highly oxidized toluene-derived SOA."* now reads "The ratio of the SOA yield on wet AS seeds to that on dry AS seeds, the relative SOA yield, decreased from 1.31±0.02 at an OH exposure of 4.66×10^10 molecules cm^-3 s to 1.01±0.01 at an OH exposure of 5.28×10^11 molecules cm^-3 s. This decrease may be due to the early deliquescence of initially dry AS seeds after coated by highly oxidized toluene-derived SOA." (Line 27-33).

*"Our results suggest that AS dry seeds soon turn to at least partially deliquesced particles during SOA formation and more studies on the interplay of SOA formation and ALW are warranted."* now reads:

"Our results suggest that inorganic dry seeds become at least partially deliquesced particles during SOA formation and hence ALW is inevitably involved in the SOA formation at moderate RH. More laboratory experiments conducted with a wide variety of SOA precursors and inorganic seeds under different $NO_x$ and RH conditions are warranted." (Line 44-48).

**Q6:** Line 36: m/z 29, 43, 44 are specific to the aerosol mass spectrometer (AMS).

R6: "obtained using an aerosol mass spectrometer (AMS)" was added. (Line 40-41).

**Q7:** Line 47-48: these references might not quite reflect the current state of knowledge. For example, Tsigaridis et al. (ACP, 2014) did a model intercomparison and found that the secondary nature of OA can be reproduced, but not the overall amount. Shrivastava et al (ACP, 2011) shows that the total SOA amount can be reproduced, but require some model tuning. Overall, there may be too many studies to cite for a simple argument here, but the de Gouw et al. and Volkamer et al. studies are quite out of date at this point.

R7: SOA models usually include update of the volatility basis set (VBS) formalism to treat gas-particle partitioning and multi-generation oxidation (Shrivastava et al., 2011; Tsigaridis et al., 2014), increased SOA yields that account for vapor wall loss in smog chambers (Zhang et al., 2014; Hayes et al., 2015) and additional SOA precursors such

as S/IVOCs (Robinson et al., 2007). These updated models can better reduce the gap between the modeled and observed SOA, but have resulted in over-prediction of SOA at long aging times. It remains unclear whether these updated models improve the simulation of SOA for the right reasons. Here, we pointed out that the presence of ALW may influence the SOA yields, which is not well treated even in the updated models. To reflect the current state of knowledge, we added the following sentence to the revised manuscript.

"The updated models incorporating the volatility basis set (VBS) formalism (Donahue et al., 2006) can better predict the observed SOA, but SOA formation still remains under-constrained (Shrivastava et al., 2011; Tsigaridis et al., 2014; Hayes et al., 2015; Ma et al., 2017)." (Line 57-60).

**Q8:** Line 60: I would also add that understanding water uptake of SOA is important for estimating its loss by wet deposition, which is highly unknown at this point.

R8: The following sentence has been added as suggested.

"In addition, understanding water uptake of SOA is important for estimating its loss by wet deposition, which is not well constrained." (Line 72-74).

**Q9:** Line 97: is the silica gel diffusion dryer manufactured by TSI? If so, consider including the model number. Is the 30% outlet RH experimentally verified? I would imagine the outlet RH would be a function of the inlet RH.

R9: It is a homemade one. The outlet RH was verified to be lower than 30%. The inlet RH should be stable during the experiment as the flow rate was stable.

**Q10:** Line 103: Is it correct that the ALW is estimated using the method described later? If so, please mention.

R10: Yes. "see Section 2.4" was added for clarification. (Line 119).

**Q11:** Lines 140-146: Based on the OH exposure calculated, what is the amount of toluene consumed?

R11: The reacted amount of toluene is now provided in Table 1. The following sentence has been added:

"The reacted and final concentrations of toluene were calculated from the OH exposure and the rate constant of the reaction between toluene and OH (Atkinson and

Arey, 2003) (Table 1)." (Line 172-174).

**Q12:** Line 166-167: is this filtered air flow with or without toluene and/or oxidation products?

R12: The filtered air flow was without oxidation products. These oxidation products were expected to have a negligible influence on the concentrations of major gases, e.g. $N_2$, $O_2$, and $CO_2$.

**Q13:** Line 177: Just to make sure that sulfate is indeed coated with SOA, it would be great to show size distributions: Either a unimodal SMPS number size distribution showing no nucleation, or the PTOF on the AMS showing that organics and sulfate are in the same mode.

R13: As shown in the following figure (now Fig. S2), at the OH exposure of $0.47\times10^{11}$ molecules $cm^{-3}$ s, the particle number distributions for both cases are unimodal, indicating no nucleation. The following sentence was added to the revised manuscript.

"The unimodal size distributions of particle numbers show the SOA formation on AS seed particles without much nucleation mode particles (Fig. S2)" (Line 198-200)

[Figure]

**Q14:** Line 187: Is it really evaporation of the organics? Would 8% mass loss due to evaporation be enough to show up in shifts in particle mode or median diameter (it would be around a ~2% change in diameter)?

R14: We cannot rule out other possibilities based on the dataset. Hence we said that it is possibly due to reversible partitioning of the SVOCs in the original text. The ~2% shift in particle mode diameter was not enough to be captured by AMS or SMPS.

**Q15:** Line 252: I suggest keeping all OH exposure numbers to 10ˆ11 molec cm-3 s. For example, I recommend changing 4.66 x 10ˆ10 to 0.47 x 10ˆ11.

R15: Revised as suggested.

**Q16:** Line 290 – 292: The authors should be cautious about observations of oligomers on the AMS. The relatively low signal above m/z 80 does not mean there is little to no oligomerization. The AMS vaporizer at 600 C causes extensive thermal decomposition.

R16: Although the AMS can cause extensive thermal decomposition, previous studies suggest that m/z > 80 can be easily observed when the oligomers are abundant (Price et al., 2014; Gilardoni et al., 2016; Faust et al., 2017). We deleted this statement due to the lack of solid evidences.

**Q17:** Line 298-303: Would a plot f(28+44) vs f(29+43) look different from f44 vs f43? The conversion of alcohols and aldehydes to acids described in the previous paragraph is fundamentally driving the trend of f44 and f43. So these two paragraphs are both conveying a similar observation, which is expected given the extensive literature on AMS description of aerosol aging. So I suggest condensing the discussion.

R17: As shown in the following figure, the f(28+44) vs f(29+43) plot looks similar as f44 vs f43 plot. The previous paragraph is focused on the comparison of the mass spectra between the lowest and highest OH exposure while this paragraph shows the overall evolution of f44 vs f43 from the lowest to the highest OH exposure, so we would like to keep the original discussions.

[Figure]

**Q18:** Line 314-315, and Fig. S4: The differences seem very minute to me and may not be statistically significant. The y-scale is misleading, and should start at zero.

R18: Fig. S4 has been revised accordingly as follows. The difference in the abundance of $C_2H_3O^+$ between dry and wet AS seeds is small but there is quite an obvious trend as OH exposure increases.

[Figure]

**Q19:** Line 314-315: If the trend is due to enhanced partitioning of water soluble organics to ALW, wouldn't the

R19: This issue of solubility of WSOC is likely addressed in response to Q1 and Q2.

**Q20:** Line 327:   It looks like the subscript "C" in OSC is not capitalized, where it has been capitalized in other instances in the manuscript.

R20: Revised.

References:

Bloss, C., Wagner, V., Jenkin, M. E., Volkamer, R., Bloss, W. J., Lee, J. D., Heard, D. E., Wirtz, K., Martin-Reviejo, M., Rea, G., Wenger, J. C., and Pilling, M. J.: Development of a detailed chemical mechanism (MCMv3.1) for the atmospheric oxidation of aromatic hydrocarbons, Atmos. Chem. Phys., 5, 641-664, 10.5194/acp-5-641-2005, 2005.

Cheung, H. H. Y., Yeung, M. C., Li, Y. J., Lee, B. P., and Chan, C. K.: Relative Humidity-Dependent HTDMA Measurements of Ambient Aerosols at the HKUST

Supersite in Hong Kong, China, Aerosol Sci Tech, 49, 643-654, 10.1080/02786826.2015.1058482, 2015.

Choi, M. Y., and Chan, C. K.: The Effects of Organic Species on the Hygroscopic Behaviors of Inorganic Aerosols, Environ Sci Technol, 36, 2422-2428, 10.1021/es0113293, 2002.

DeCarlo, P. F., Dunlea, E. J., Kimmel, J. R., Aiken, A. C., Sueper, D., Crounse, J., Wennberg, P. O., Emmons, L., Shinozuka, Y., Clarke, A., Zhou, J., Tomlinson, J., Collins, D. R., Knapp, D., Weinheimer, A. J., Montzka, D. D., Campos, T., and Jimenez, J. L.: Fast airborne aerosol size and chemistry measurements above Mexico City and Central Mexico during the MILAGRO campaign, Atmos. Chem. Phys., 8, 4027-4048, 10.5194/acp-8-4027-2008, 2008.

Donahue, N. M., Robinson, A. L., Stanier, C. O., and Pandis, S. N.: Coupled Partitioning, Dilution, and Chemical Aging of Semivolatile Organics, Environ Sci Technol, 40, 2635-2643, 10.1021/es052297c, 2006.

Faust, J. A., Wong, J. P. S., Lee, A. K. Y., and Abbatt, J. P. D.: Role of Aerosol Liquid Water in Secondary Organic Aerosol Formation from Volatile Organic Compounds, Environ Sci Technol, 51, 1405-1413, 10.1021/acs.est.6b04700, 2017.

George, I. J., and Abbatt, J. P. D.: Heterogeneous oxidation of atmospheric aerosol particles by gas-phase radicals, Nat Chem, 2, 713-722, 2010.

Gilardoni, S., Massoli, P., Paglione, M., Giulianelli, L., Carbone, C., Rinaldi, M., Decesari, S., Sandrini, S., Costabile, F., Gobbi, G. P., Pietrogrande, M. C., Visentin, M., Scotto, F., Fuzzi, S., and Facchini, M. C.: Direct observation of aqueous secondary organic aerosol from biomass-burning emissions, Proceedings of the National Academy of Sciences, 113, 10013-10018, 10.1073/pnas.1602212113, 2016.

Hayes, P. L., Carlton, A. G., Baker, K. R., Ahmadov, R., Washenfelder, R. A., Alvarez, S., Rappenglück, B., Gilman, J. B., Kuster, W. C., de Gouw, J. A., Zotter, P., Prévôt, A. S. H., Szidat, S., Kleindienst, T. E., Offenberg, J. H., Ma, P. K., and Jimenez, J. L.: Modeling the formation and aging of secondary organic aerosols in Los Angeles during CalNex 2010, Atmos. Chem. Phys., 15, 5773-5801, 10.5194/acp-15-5773-2015, 2015.

Ip, H. S. S., Huang, X. H. H., and Yu, J. Z.: Effective Henry's law constants of glyoxal,

glyoxylic acid, and glycolic acid, Geophys Res Lett, 36, L01802, 10.1029/2008GL036212, 2009.

Jenkin, M. E., Saunders, S. M., Wagner, V., and Pilling, M. J.: Protocol for the development of the Master Chemical Mechanism, MCM v3 (Part B): tropospheric degradation of aromatic volatile organic compounds, Atmos. Chem. Phys., 3, 181-193, 10.5194/acp-3-181-2003, 2003.

Kampf, C. J., Waxman, E. M., Slowik, J. G., Dommen, J., Pfaffenberger, L., Praplan, A. P., Prévôt, A. S. H., Baltensperger, U., Hoffmann, T., and Volkamer, R.: Effective Henry's Law Partitioning and the Salting Constant of Glyoxal in Aerosols Containing Sulfate, Environ Sci Technol, 47, 4236-4244, 10.1021/es400083d, 2013.

Liggio, J., Li, S.-M., and McLaren, R.: Reactive uptake of glyoxal by particulate matter, Journal of Geophysical Research: Atmospheres, 110, D10304, 10.1029/2004JD005113, 2005.

Ma, P. K., Zhao, Y., Robinson, A. L., Worton, D. R., Goldstein, A. H., Ortega, A. M., Jimenez, J. L., Zotter, P., Prévôt, A. S. H., Szidat, S., and Hayes, P. L.: Evaluating the impact of new observational constraints on P-S/IVOC emissions, multi-generation oxidation, and chamber wall losses on SOA modeling for Los Angeles, CA, Atmos. Chem. Phys., 17, 9237-9259, 10.5194/acp-17-9237-2017, 2017.

Nguyen, T. K. V., Zhang, Q., Jimenez, J. L., Pike, M., and Carlton, A. G.: Liquid Water: Ubiquitous Contributor to Aerosol Mass, Environmental Science & Technology Letters, 3, 257-263, 10.1021/acs.estlett.6b00167, 2016.

Peng, Z., Day, D. A., Ortega, A. M., Palm, B. B., Hu, W., Stark, H., Li, R., Tsigaridis, K., Brune, W. H., and Jimenez, J. L.: Non-OH chemistry in oxidation flow reactors for the study of atmospheric chemistry systematically examined by modeling, Atmos. Chem. Phys., 16, 4283-4305, 10.5194/acp-16-4283-2016, 2016.

Price, D. J., Clark, C. H., Tang, X., Cocker, D. R., Purvis-Roberts, K. L., and Silva, P. J.: Proposed chemical mechanisms leading to secondary organic aerosol in the reactions of aliphatic amines with hydroxyl and nitrate radicals, Atmos Environ, 96, 135-144, https://doi.org/10.1016/j.atmosenv.2014.07.035, 2014.

Robinson, A. L., Donahue, N. M., Shrivastava, M. K., Weitkamp, E. A., Sage, A. M.,

Grieshop, A. P., Lane, T. E., Pierce, J. R., and Pandis, S. N.: Rethinking Organic Aerosols: Semivolatile Emissions and Photochemical Aging, Science, 315, 1259-1262, 10.1126/science.1133061, 2007.

Shrivastava, M., Fast, J., Easter, R., Gustafson Jr, W. I., Zaveri, R. A., Jimenez, J. L., Saide, P., and Hodzic, A.: Modeling organic aerosols in a megacity: comparison of simple and complex representations of the volatility basis set approach, Atmos. Chem. Phys., 11, 6639-6662, 10.5194/acp-11-6639-2011, 2011.

Svenningsson, B., Rissler, J., Swietlicki, E., Mircea, M., Bilde, M., Facchini, M. C., Decesari, S., Fuzzi, S., Zhou, J., Mønster, J., and Rosenørn, T.: Hygroscopic growth and critical supersaturations for mixed aerosol particles of inorganic and organic compounds of atmospheric relevance, Atmos. Chem. Phys., 6, 1937-1952, 10.5194/acp-6-1937-2006, 2006.

Tsigaridis, K., Daskalakis, N., Kanakidou, M., Adams, P. J., Artaxo, P., Bahadur, R., Balkanski, Y., Bauer, S. E., Bellouin, N., Benedetti, A., Bergman, T., Berntsen, T. K., Beukes, J. P., Bian, H., Carslaw, K. S., Chin, M., Curci, G., Diehl, T., Easter, R. C., Ghan, S. J., Gong, S. L., Hodzic, A., Hoyle, C. R., Iversen, T., Jathar, S., Jimenez, J. L., Kaiser, J. W., Kirkevåg, A., Koch, D., Kokkola, H., Lee, Y. H., Lin, G., Liu, X., Luo, G., Ma, X., Mann, G. W., Mihalopoulos, N., Morcrette, J. J., Müller, J. F., Myhre, G., Myriokefalitakis, S., Ng, N. L., O'Donnell, D., Penner, J. E., Pozzoli, L., Pringle, K. J., Russell, L. M., Schulz, M., Sciare, J., Seland, Ø., Shindell, D. T., Sillman, S., Skeie, R. B., Spracklen, D., Stavrakou, T., Steenrod, S. D., Takemura, T., Tiitta, P., Tilmes, S., Tost, H., van Noije, T., van Zyl, P. G., von Salzen, K., Yu, F., Wang, Z., Wang, Z., Zaveri, R. A., Zhang, H., Zhang, K., Zhang, Q., and Zhang, X.: The AeroCom evaluation and intercomparison of organic aerosol in global models, Atmos. Chem. Phys., 14, 10845-10895, 10.5194/acp-14-10845-2014, 2014.

Wong, J. P. S., Lee, A. K. Y., and Abbatt, J. P. D.: Impacts of Sulfate Seed Acidity and Water Content on Isoprene Secondary Organic Aerosol Formation, Environ Sci Technol, 49, 13215-13221, 10.1021/acs.est.5b02686, 2015.

Zhang, X., Cappa, C. D., Jathar, S. H., McVay, R. C., Ensberg, J. J., Kleeman, M. J., and Seinfeld, J. H.: Influence of vapor wall loss in laboratory chambers on yields of

secondary organic aerosol, P. Natl. Acad. Sci., 111, 5802-5807, 10.1073/pnas.1404727111, 2014.